# Time-resolved X-ray imaging of the current filamentation instability in solid-density plasmas

Christopher Schoenwaelder [1,2] ✉, Alexis Marret[1], Stefan Assenbaum [3,4], Chandra B. Curry [1,5], Eric Cunningham [6], Gilliss Dyer [6], Stefan Funk[2], Griffin D. Glenn [1,7], Sebastian Goede[8], Dimitri Khaghani [6], Martin Rehwald [3], Ulrich Schramm [3], Franziska Treffert [1,9], Milenko Vescovi [3,4], Karl Zeil [3], Siegfried H. Glenzer [1], Frederico Fiuza [1,10] ✉ & Maxence Gauthier [1] ✉

The streaming of energetic charged particles can magnetize astrophysical and laboratory plasmas via the current filamentation instability. Despite its importance, the experimental characterization of this instability has remained a challenge. Here, we report an experiment combining a high-intensity optical laser with a high-brightness X-ray free electron laser that successfully images the instability in solid-density plasmas with 200 nm spatial and 50 fs temporal resolution. We characterize the development of µm-scale filamentary structures and their evolution over tens of picoseconds through a non-linear merging process. The measured plasma density modulations and long merging time reveal the critical importance of space-charge effects and ion motion on this electron-driven instability. Supporting theoretical analysis and kinetic simulations help distinguish the relative role of space-charge and resistive effects. Our findings indicate that magnetic fields on the order of 10 megagauss are produced, with important implications for transport and radiation emission of energetic particles in plasmas.

It has long been known that the propagation of high-energy charged particles in plasmas drives kinetic instabilities[1] that produce growing electrostatic and electromagnetic fields. These (collective) fields control plasma dynamics and their study is critical to a wide range of systems, from cosmic-ray transport in galaxies[2] to the solar wind[3] to fusion plasmas[4]. Of primary interest is the so-called Weibel[5], or current filamentation instability[6], which plays an important role in the self-magnetization of many astrophysical and laboratory settings. These include the amplification of magnetic fields in young supernova remnant shocks[7–9] and gamma-ray bursts[10–12], which shape particle acceleration and radiation emission from these environments. This instability is also seen as a key candidate for the generation of the seed fields[13] required to explain the ubiquitous galactic magnetic fields via a dynamo model[14].

The study of the current filamentation instability, however, is challenged by its rapid growth and short (kinetic) spatial scale, which prevents its direct identification and characterization from astrophysical observations. Over the past decades, this has motivated

[1]High-Energy Density Science Division, SLAC National Accelerator Laboratory, Menlo Park, CA, USA. [2]Erlangen Centre for Astroparticle Physics, Friedrich-Alexander University Erlangen-Nuremberg, Erlangen, Germany. [3]Institute of Radiation Physics, Helmholtz-Zentrum Dresden-Rossendorf, Dresden, Germany. [4]TUD Dresden University of Technology, Dresden, Germany. [5]Department of Electrical and Computer Engineering, University of Alberta, Edmonton, AB, Canada. [6]Linac Coherent Light Source, SLAC National Accelerator Laboratory, Menlo Park, CA, USA. [7]Applied Physics Department, Stanford University, Stanford, CA, USA. [8]European XFEL, Schenefeld, Germany. [9]Institut für Kernphysik, Technische Unversität Darmstadt, Darmstadt, Germany. [10]GAP/Instituto de Plasmas e Fusão Nuclear, Instituto Superior Técnico, Universidade de Lisboa, Lisbon, Portugal. ✉e-mail: schchris@slac.stanford.edu; frederico.fiuza@tecnico.ulisboa.pt; gauthier@slac.stanford.edu

significant interest in controlled laboratory studies using various experimental configurations, including electron beams from conventional and laser-driven particle accelerators[15,16], field-ionized gases[17], and non-relativistic colliding plasma flows driven by long (nanosecond) laser pulses[7,18]. In addition, there has been a substantial effort in investigating this instability in high-intensity laser-solid interactions, which are at the core of numerous cross-field applications, from advanced inertial fusion schemes[4,19,20] to compact plasma-based accelerators[21,22]. In this context, the instability is central to understanding energetic electron transport and energy deposition, plasma heating, and magnetic field generation.

The high flux of relativistic (hot) electrons produced by an intense laser in a solid-density plasma generates a mega-ampere current[23] and drives a return current in the background plasma. The resulting counter-propagating electron populations become unstable to the current filamentation instability and generate a magnetic field[4,24]. The strong density and temperature asymmetry between the two electron populations poses significant challenges to theoretical analysis; most theoretical and numerical studies have focused on the simplest symmetric case, where the instability is purely electromagnetic[11,12]. The few existing studies of the asymmetric case suggest that space-charge effects can play an important role[25–27]. Resistive effects on the dense background plasma further complicate the description of the instability[27]. The competition between these effects in the linear phase of the instability remains unclear. The nonlinear evolution—critical to understanding the dynamics of the resulting magnetic field and its implications for the transport of energetic particles—is even less understood, highlighting the need for experimental characterization.

Previous experiments have primarily relied on optical probing[24,28], as well as laser-driven electron[29–31] and proton radiography[21,32–34] of the laser-solid interaction to diagnose the current filamentation instability; however, these studies were limited to probing either far from the interaction region[24,34] or the low-density plasma outside the solid target[28,32]. This is because optical radiation cannot penetrate the dense plasma and charged particles will be deflected by the strong electromagnetic fields generated at the target surface, preventing or corrupting the measurement within the interaction region. Moreover, kinetic scale processes, such as those associated with the filamentation instability at solid density, are particularly challenging to investigate,

as they require very high spatial resolution (typically, sub-$\mu$m). While laser-driven X-ray sources can penetrate the dense plasma, they cannot yet deliver the required spatial resolution, narrow bandwidth and high flux needed to characterize the growth of this instability[35–37].

Here, we take advantage of the high-brightness and narrow-bandwidth Linac Coherent Light Source (LCLS) XFEL to overcome these limitations in an X-ray phase contrast microscopy configuration, enabling an unprecedented characterization of the filamentation instability in solid-density plasmas. The resulting high-resolution images show the clear development of filamentary density modulations over picosecond time scales and reveal, as we will show, that space-charge effects and ion motion play an important role in significantly slowing down the growth of the instability and that strong, 10 MG-level, magnetic fields are produced.

## Results and discussion
### Experimental overview
Our experiment was conducted at the Matter in Extreme Conditions (MEC) end-station of the SLAC National Accelerator Laboratory, in an optical-pump X-ray-probe configuration, as shown in Fig. 1. The high-contrast MEC short-pulse laser (0.8 $\mu$m wavelength, 1 J energy, 150 fs pulse duration, P-polarization)[38] is focused on $\varnothing$ 10 $\mu$m copper and $\varnothing$ 15 $\mu$m aluminum wire targets with a focal spot of ~6 $\mu$m, reaching an intensity of $\approx 1.4 \times 10^{19}$ W cm$^{-2}$. The high peak-brightness Linac Coherent Light Source (LCLS) X-ray laser (650 $\mu$J energy, 50 fs pulse duration) probes the laser-target interaction and subsequent plasma dynamics at 80° w.r.t. the pump-laser axis in an X-ray phase contrast microscopy configuration (see "Methods").

By varying the timing between the optical pump and X-ray probe lasers from $\Delta t = 0.5$ ps to 800 ps, we identify three distinct phases of the plasma evolution. At early times ($\leq$40 ps), we observe the development of clear filamentary structures, whose kinetic scale and transverse modulation are consistent with the current filamentation instability. At larger delays, the hydrodynamic evolution is revealed, with the observation of a laser-driven shock wave traveling at ~85 km s$^{-1}$, and the thermal expansion of the heated target region. In the following, we will focus on the study of filamentary structures and associated plasma instabilities observed within the target, corresponding to $\Delta t \leq 40$ ps.

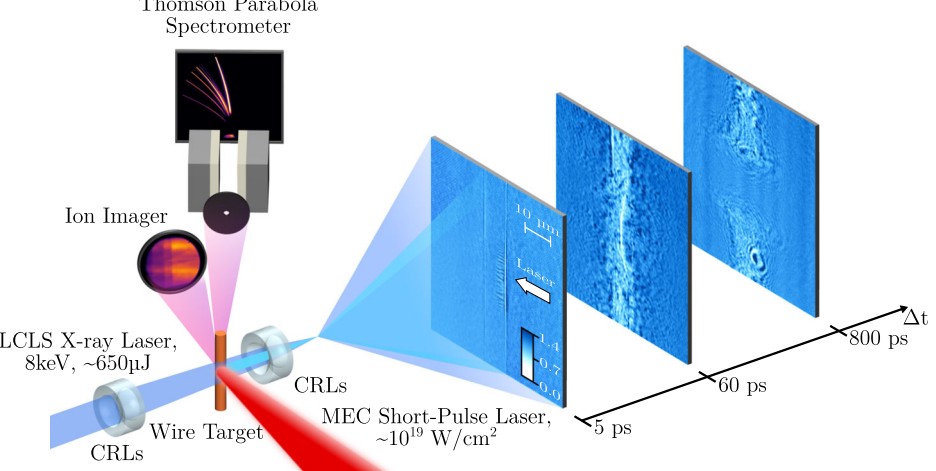

**Fig. 1 | Experimental setup for ultra-fast imaging of the laser-target interaction.** The 1 J optical pump laser (red) is focused onto $\varnothing$ 10 $\mu$m copper and $\varnothing$ 15 $\mu$m aluminum wires, generating a laser intensity on target of 1.4 × 10$^{19}$ W cm$^{-2}$. The X-ray laser (blue) probes the interaction at 80° with respect to the pump-laser axis and is imaged onto an X-ray camera using Compound Refractive Lenses positioned after the target. Three phases of the laser-target interaction are observed. Images at short pump-probe delays (<40 ps) reveal the development of density filaments along the direction of the pump-laser axis. At larger delays, we observe the hydrodynamic evolution of the target, including a laser generated shock and the subsequent thermal expansion. The colorbar indicates the intensity of the signal normalized by either a reference image of the wire before shot (5 ps) or the X-ray background (60 ps and 800 ps). Additional diagnostics include a Thomson Parabola Spectrometer, Ion Imager, and Transmission Diagnostic (not depicted in the figure).

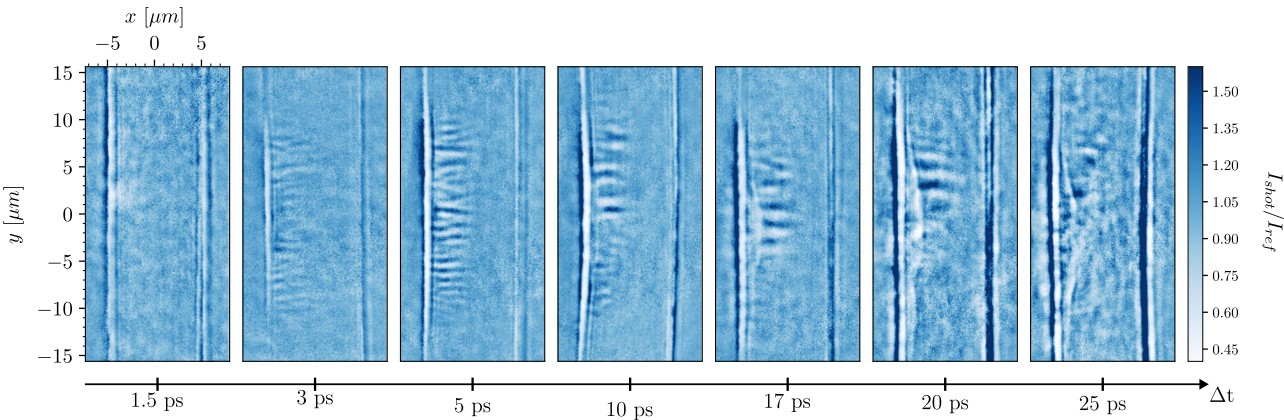

**Fig. 2 | X-ray images of the high-intensity laser interaction with a $\varnothing\,10\,\mu m$ copper wire at varying pump-probe delays $\Delta t$ visualize the temporal evolution of the current filamentation instability inside the resulting solid-density** **plasma.** The transmission signal $I_{shot}$ is normalized with a reference signal $I_{ref}$ of the wire before the laser interaction. Similar images were obtained for $\varnothing\,15\,\mu m$ aluminum wire targets. The images present only a subset of the full data set.

## Hot electron temperature

Upon interacting with the solid density target, the high-intensity optical laser produces a small fraction of very energetic (hot) electrons[39,40], with density $n_h \approx n_{crit}$, that propagate in the resulting plasma of density $n_{e0} \gg n_{crit}$, where $n_{crit}$ is the critical density for the propagation of the laser in the plasma. As the hot electrons reach the rear side of the target, a small fraction escapes and establishes a strong electrostatic sheath field, while the majority is reflected and recirculates within the target[41]. The sheath field accelerates rear surface ions via the Target Normal Sheath Acceleration (TNSA) process[42]. The energy spectrum of these energetic ions is measured via a Thomson Parabola Spectrometer (TPS), and can be modeled as $dN/d\mathcal{E} \propto \exp(-\sqrt{2\mathcal{E}/T_h})/\sqrt{\mathcal{E}}$, where $\mathcal{E}$ is the particle energy[43] (see "Methods"). Following this model, we can infer the slope temperature $T_h$ of the hot electrons, which is measured to be $T_h = 400 \pm 30$ keV and $700 \pm 40$ keV for the aluminum and copper targets, respectively. This is in good agreement with laser-induced electron heating models for high-contrast laser-solid interactions[39,40] that predict hot electron temperatures in the range $T_h = 500$–$900$ keV for the MEC laser conditions (see "Methods").

## Current filamentation instability

The hot electrons drive a return current in the target as they propagate. The resulting counter-propagating electron populations then trigger the current filamentation instability. The associated plasma density evolution is captured through a series of single-shot images. These snapshots provide a spatial resolution of $\approx 200$ nm and a temporal resolution determined by the 50 fs X-ray pulse duration, as well as the 100 fs precision of setting the pump-probe delay. High sensitivity to density variations is achieved via the novel MEC X-ray Imager (MXI) diagnostic[44] that combines beryllium Compound Refractive Lenses (CRLs)[45] positioned behind the target with an X-ray Camera. Due to the chromaticity of the CRLs and the bandwidth of the X-ray laser, both phase and absorption effects contribute to the X-ray signal, resulting in a high-resolution image with strong sensitivity to phase contrast effects introduced by density gradients inside the target.

The resulting X-ray images, displayed in Fig. 2, show the formation of density filaments with an initial wavelength $\lambda \approx 1\,\mu m$, evolving from $\Delta t = 1.5$–25 ps in the copper wire targets. Similar images are obtained for the aluminum targets, exhibiting filaments with an initial wavelength comparable to those in copper. We note that the distribution of wavelength components in the 3D structure is conserved through its 2D projection[46], as well as the imaging system (see "Methods"). The laser-irradiated target surface remains very sharp during the growth of the filaments; the hydrodynamic evolution of the front surface only

starts to be noticeable between 10 and 20 ps after the laser irradiation (see Fig. 2). This, together with the high-contrast of the X-ray images, clearly indicates that the filaments are produced in the solid-density plasma.

In the case of purely electromagnetic perturbations, both counter-propagating electron populations would be deflected at the same rate, producing symmetric filaments of alternating current polarity and no modulations in the total density. Since the MXI diagnostic is predominantly sensitive to gradients in electron density, such filaments would not produce a measurable signal. The alternating regions of high and low X-ray transmission, instead, show an asymmetric electron density distribution between filaments with opposite polarities, demonstrating that space-charge effects and ion motion must play an important role in the development of the instability in solid-density plasmas. This asymmetry arises because the hot and background electrons, having vastly different densities and temperatures, are deflected at different rates in the self-generated magnetic fields[25]. The resulting electron density perturbations produce electrostatic fields that set in motion the ions transversely to maintain quasi-neutrality. The plasma filamentation thus involves both electrons and ions even if the instability is triggered by the laser-produced fast electrons.

The evolution of the filament wavelength, shown in Fig. 3, is measured using lineouts of the X-ray images taken at different pump-probe delays (see "Methods"). The wavelength increases up to $2\,\mu m$ for copper and $3\,\mu m$ for aluminum over $\approx 20$ ps, as a result of the noninear evolution of the current filamentation instability.

Current filaments of the same polarity attract each other via the magnetic force, giving rise to an inverse cascade as they merge and increase in size[47,48]. This merging process can be modeled by considering a distribution of identical filaments with an average initial current $I_0$, radius $R_0$, and separation of $2R_0$. The magnetic field associated with each filament is $B_0 = 2I_0/(cr)$, with $r$ the cylindrical radius. The resulting attractive force per unit length acting on a neighboring filament of the same polarity is $dF/dl = -B_0I_0/c$. Following ref. 47 the wavelength evolution $\lambda(t)$ is dictated by the pair-wise self-similar merging dynamics of the filaments, given by

$$\lambda(t) = \lambda_0 2^{\frac{t}{2\tau_m}}, \tag{1}$$

with

$$\tau_m \approx 0.39(mn)^{1/2}\frac{\lambda_0}{B_0} \tag{2}$$

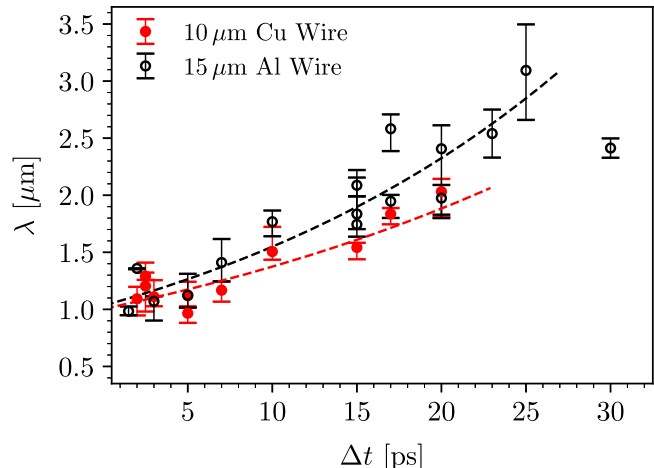

**Fig. 3 | Temporal evolution of the filament wavelength in $\varnothing$ 10 $\mu$ m copper (red solid circles) and $\varnothing$ 15 $\mu$ m aluminum (black open circles) wire targets.** Each data point represents a single shot. The range markers indicate the first and third quartile of the wavelength distribution at different pump-probe delays $\Delta t$. The dotted lines represent the best fits to the filament merging model according to Eq. (1). The average error and standard deviation of the distributions (not shown in the figure) are 0.15 $\mu$m and 0.02 $\mu$m for the copper measurements and 0.19 $\mu$m and 0.06 $\mu$m for aluminum, respectively.

the characteristic merging time of two neighboring filaments, $\lambda_0 = 4R_0$ the wavelength after saturation of the instability (end of linear phase), $m$ the mass of the particles in the filament and $n$ their number density. It is clear that the merging of filaments in the asymmetric regime cannot involve only the electron dynamics, as typically considered in the purely electromagnetic case[47]. Instead, the mass density carried during the merging process is dominated by that of the ions $mn \approx m_i n_i$ to maintain quasi-neutrality, even if the current is carried by the electrons. This considerably increases the merging time of the filaments in the non-linear phase by a factor $[m_i/(Z m_e)]^{1/2}$ where $Z$ is the ionization level.

We find that this merging model can provide a good description of the observed wavelength growth, as shown in Fig. 3. From fitting Eq. (1) to the wavelength measurements, the extracted merging time is $\tau_m \approx 11$ ps and $\tau_m \approx 8$ ps for copper and aluminum, respectively. From Eq. (2) the inferred magnetic field associated with the filaments is $B_0 \approx 10$ MG for copper, and $B_0 \approx 7$ MG for aluminum (see "Methods"). Such strong magnetic fields are consistent with the Alfvén limit for the fast electron current[49]. Considering that for a hot electron temperature of a few hundred keV, the typical drift velocity in the laser propagation direction is $v_{dh} \approx 0.5 c$, the electron Larmor radius for $B_0 = 10$ MG is 1 $\mu$m, which corresponds to the filament wavelength measured in the experiment. Once such large magnetic fields are produced in the plasma, they will affect fast electron transport, helping confined them within the filaments.

At later times $\Delta t > 20$ ps, the filaments start to rapidly decay. Interestingly, this is comparable to the predicted stopping range of hot electrons in the dense plasma. For a hot electron temperature $T_h = 700$ keV, and typical values for the background plasma temperature $T_e = 100$ eV and density $n_{e0} = 5.8 \times 10^{23}$ cm$^{-3}$ [50,51] (assuming a copper target with $Z = 7$), the stopping distance is $L_s \approx 0.35$ cm[52]. For $v_{dh} \approx 0.5 c$, this corresponds to a stopping time of $\tau_s \approx 23$ ps. Future experiments combining X-ray imaging with X-ray Thomson Scattering or K-alpha probes could help confirm whether the life time of the filaments is indeed determined by the hot electron stopping time and how the generated magnetic fields affect the fast electron transport. We note that at such late times the filament detection is further complicated by the propagation of the laser-driven shock into the

target and the increased background noise level, as can be observed in Fig. 2. Additionally, the MXI diagnostic becomes decreasingly sensitive to the filaments as their electron density gradient reduces with the growing wavelength.

The experimental results reveal the importance of electrostatic effects and ion motion in the dynamics of the asymmetric current filamentation instability produced in solid-density plasmas. Furthermore, they suggest that strong magnetic fields are produced, capable of affecting fast electron transport and confinement. We now compare these findings with fully kinetic simulations and analytic theory to further elucidate the interplay between these effects in the solid density plasma.

**Kinetic simulations and theoretical analysis.** We performed simulations of the laser-plasma interaction with the OSIRIS 4.0 particle-in-cell (PIC) code[53]. The laser irradiation of the copper wire and subsequent plasma evolution are described fully kinetically, including binary Coulomb collisions while maintaining a realistic plasma density and ion to electron mass ratio. The two-dimensional simulation domain models a lateral view of the laser-solid interaction, with a box size of $40 \times 40$ $\mu$m, centered on the $\varnothing$ 10 $\mu$m copper wire (see "Methods"). The laser parameters are chosen to match the experimental conditions. Due to the high computational expense of kinetic simulations we focus on the first 3 ps of the interaction, which allows us to investigate the linear phase of growth and saturation of the filamentation instability. In particular, we focus our analysis on the microphysics governing the filament growth observed in the experiment and on the interplay between resistive and space-charge effects.

We first address the role of space-charge effects and ion dynamics by modeling the interaction without Coulomb collisions. As the laser-driven hot electrons propagate and recirculate in the plasma, we clearly observe the development of density filaments with a growth rate $\Gamma \approx 0.5$ ps$^{-1}$ and a dominant wavelength of $\lambda \approx 0.12$ $\mu$m, as seen in Fig. 4a. During the growth of the instability we measure a characteristic hot electron slope temperature of $T_h \approx 400$ keV along the laser propagation direction and 100 keV transversely, an average drift velocity $v_{dh} \approx 0.3 c$, and a density $n_h \approx 0.02$ $n_{e0}$. The large difference in temperature and density between the hot electrons and the cold return current populations drives an asymmetric instability characterized by total density modulations with $\delta n/n \approx 5$ %, consistent with the experimental observations. We verified that the large hot electron temperature transverse to the laser propagation direction renders this population stable to the collisionless current filamentation instability[4], further confirming the importance of the cold return current. When the same simulation is repeated with ion motion switched off—that is, ions are modeled as an infinitely massive charge-neutralizing background species—density modulations are no longer visible. This highlights the critical role of space-charge effects in the development of the large density perturbations observed in both the simulations and experiment.

The simulations results are supported by theoretical calculations of the dispersion relation[25]. Figure 4d shows a comparison of the collisionless theoretical growth rate with and without coupling to electrostatic modes for the relevant hot electron and target parameters (see "Methods"). Space-charge effects are found to reduce the instability growth rate by more than an order of magnitude in comparison to the purely electromagnetic case, leading to a predicted growth rate $\Gamma \approx 0.6$ ps$^{-1}$ and most unstable wavelength $\lambda \approx 0.2$ $\mu$m, in agreement with the collisionless simulations.

We then assess the effects of target resistivity by comparing the simulation results without and with Coulomb collisions, as illustrated in Fig. 4a, b. We find that in the resistive case the growth rate is increased to $\Gamma \approx 0.8$ ps$^{-1}$ with the most unstable mode shifted towards larger wavelength, leading to $\lambda \approx 0.7$ $\mu$m, consistent with the experimental observations (see Fig. 4d, green and red markers). This

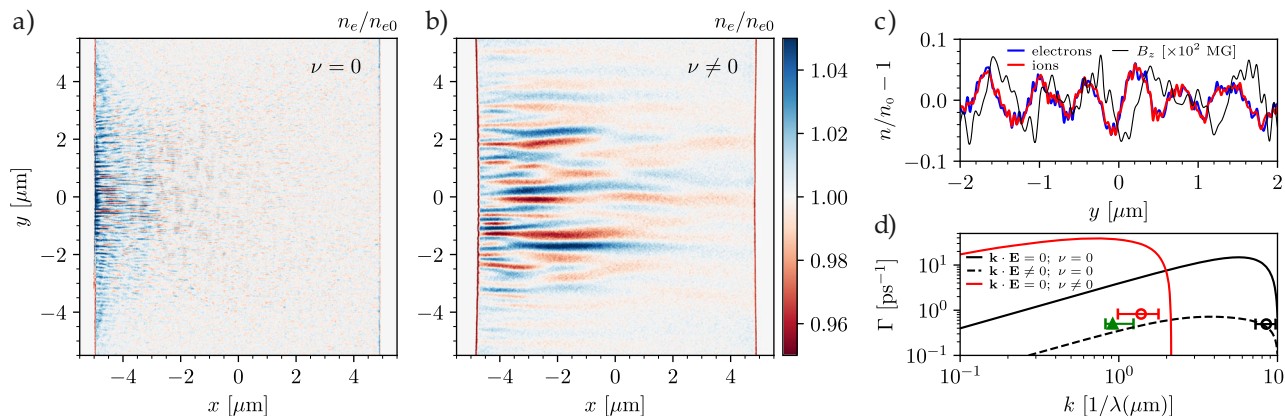

**Fig. 4 | Particle-in-cell simulation results of the laser-copper plasma interaction. a, b** show normalized electron density maps at $t = 2$ ps after the laser peak intensity reaches the target surface for the collisionless ($v = 0$) and collisional/resistive ($v \neq 0$) cases. The laser comes in from the left-hand side. **c** Lineout of the normalized electron (blue) and ion (red) density, and transverse magnetic field (black), taken at $t = 2$ ps and $x = -3$ μm, $y \in [-2, 2]$ μm for the collisional simulation. **d** Linear theory prediction of the growth rate $\Gamma$ as a function of wavenumber $k$ in the purely electromagnetic collisionless case ($\mathbf{k} \cdot \mathbf{E} = 0$; $v = 0$, solid black), including electrostatic effects (dashed black), and purely electromagnetic resistive case (solid red). The red and black circles indicate the simulations results in the resistive (red) and collisionless (black) cases respectively, the range markers indicate the measured distribution of dominant wavenumbers. The green marker indicates the experimental measurement, where the growth rate is approximated as the inverse of the earliest time filaments are visible in the X-ray images ($\Delta t = 2$ ps) and $k$ is the corresponding median wavenumber. The associated range markers indicate the distribution of wavenumbers observed.

migration towards larger wavelength, associated with the increased heating of the background electrons in the resistive case[27], is predicted by theoretical calculations for the purely electromagnetic resistive current filamentation instability, as shown in Fig. 4d. A transverse lineout of the electron and ion density is shown in Fig. 4c, taken 3 μm away from the target surface in the collisional simulation. The ions are found to filament together with the electrons and produce density modulations with a similar amplitude when compared to the collisionless simulation, demonstrating the importance of ion dynamics in both the resistive and collisionless cases. The filament growth is accompanied by strong amplification of the magnetic field reaching $B \approx 7$ MG (Fig. 4c, black curve). This is again consistent with the Alfvén limit for the hot electron drift velocity $v_{dh} \approx 0.3\,c$ and filament wavelength $\lambda = 0.7$ μm measured in the simulations, and in good agreement with the values inferred from the filament merging rate in the experiment.

Our results reveal the combined importance of space-charge and resistive effects on the growth and nonlinear dynamics of the current filamentation instability, which significantly slow down its growth rate and increase the dominant wavelength. The strong, 10 MG level magnetic fields produced by the instability can impact various applications, potentially enabling high brilliance synchrotron radiation sources[54,55] and improving the collimation of fast electrons in fast ignition inertial confinement fusion schemes. More generally, this work opens up a new experimental route for detailed studies of the microphysics of relativistic streaming instabilities of relevance to both laboratory and astrophysical plasmas[12,56]. Future extensions to currently available higher laser intensity and lower density targets[21] could probe deeper nonlinear regimes and the mechanisms behind the formation of relativistic shock waves relevant to gamma-ray bursts[10,12]. Furthermore, at higher laser intensities, even larger magnetic fields will be produced, potentially enabling direct measurements of density and magnetic field structures by combining X-ray imaging with advances in X-ray polarimetry[57].

## Methods
### Experimental setup
The experiment presented in this work was conducted at the SLAC National Accelerator Laboratory. At the Matter in Extreme Conditions (MEC) endstation we combined the MEC high-intensity optical short-

pulse laser with the Linac Coherent Light Source (LCLS) high peak-brightness X-ray laser in a pump-probe configuration. The 1 J, 150 fs, 800 nm Ti:Sapphire high intensity contrast (>$10^7$ at t = −3 ps)[38] MEC pump laser (P-polarization) was focused on 10 μm copper and 15 μm aluminum wire targets via an f/6 Off-Axis Parabola (OAP). The focal spot is of 5 μm FWHM along the wire axis direction and 7 μm along the transverse direction, containing 60% of the laser energy and generating intensities on target of $\approx 1.4 \times 10^{19}$ W cm$^{-2}$. The 8 keV, 650 μJ, 50 fs LCLS X-ray probe laser was pre-focused to a beam diameter of $\varnothing 150\,\mu$ m via upstream CRLs to back illuminate the laser-target interaction at an angle of 80° w.r.t. the optical pump laser.

The attenuated X-ray beam was imaged onto an Andor Neo 5.5 sCMOS camera via an additional set of CRLs placed after the target to achieve a spatial resolution of $\approx 200$ nm. Due to the chromaticity of the beryllium CRLs and the bandwidth of the LCLS beam (~20 eV), the imaging system possesses a focal range instead of a single focal point. This leads to an effective propagation distance $d = 0.8$–1 mm for the X-rays, which ensures that both absorption and phase effects contribute to the obtained images. The MEC time tool allows for precise measurement of the delay in arrival between the pump and probe beams, with a range of 100 fs. A leakage of the optical laser upstream is used to image the wire target in three microscopy imaging systems to aid the alignment process and increase the shot rate. A Transmission diagnostic captures the transmitted light of the interaction to deliver a measure of the interaction quality. A Thomson Parabola Spectrometer and Ion Imager measure the accelerated ion beam energy spectrum and spatial profile. Additionally, the combination of these secondary diagnostics allows us to quantify the quality of the interaction on shot and deliver dismissal criteria for the X-ray images. Any measurements with a large laser transmission and simultaneous poor flux and ion cut-off energy measured on the Ion Imager and Thomson Parabola Spectrometer were discarded from the analysis.

### Laser contrast and pre-plasma formation
The temporal contrast of the MEC Ti: Sapphire laser was characterized using a Tundra autocorrelator, showing excellent performance up to the main pulse[38]. The intensity contrast remains better than $10^{-9}$ up to −5 ps and better than $10^{-7}$ beyond that. To estimate the preplasma formation due to this contrast profile, we consider that significant laser energy absorption into solid copper begins above an intensity of

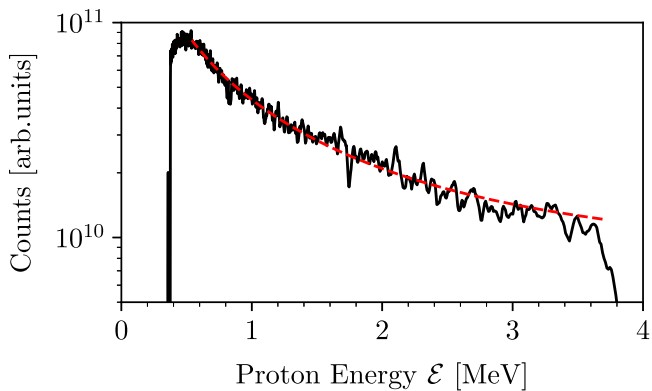

**Fig. 5 | Example of a proton spectrum as measured by the Thomson parabola spectrometer to extract the hot electron temperature $T_h$.** The raw spectrum (black) is fitted with the function shown in Eq. (3) (red, dashed).

$-10^{12}$ W cm$^{-2}$ [58]. This threshold is reached approximately at $-3$ ps for the MEC laser contrast curve. By integrating the laser intensity from $-3$ to $-0.5$ ps and applying intensity-dependent absorption coefficients from the same reference, we estimate that $-2\,\mu$J of energy is deposited in the target prior to the main pulse. Assuming the energy is deposited within a 25.8 nm thick surface layer[59] over a $-5\,\mu$m diameter spot, and using copper's atomic density with partial ionization ($Z = 4-5$), we estimate an electron temperature of 30–50 eV. This yields a plasma expansion velocity of $- 0.02\,\mu$m ps$^{-1}$. Given the short time between the onset of the pre-pulse and the arrival of the main pulse, the plasma does not expand significantly. As a result, the estimated preplasma scalelength at the time of interaction remains below 0.1 $\mu$m, allowing the laser to interact with a steep density gradient near the original solid surface.

## Ion energy spectrum and hot electron temperature

The Thomson parabola spectrometer was set at a 36° angle w.r.t. the laser axis to measure the spectrum of energetic ions generated during the interaction. A low divergence portion of the beam is extracted using a $\varnothing$ 130 $\mu$m pinhole at a distance of 1.315 m from the target, corresponding to an acceptance angle of 0.77 $\mu$sr. The ions are then deflected by a 0.525 T magnetic field to distinguish particles of different energy along one direction, and a 6 kV cm$^{-1}$ electric field to differentiate particles with different charge-to-mass ratio along the other direction. The particles are visualized by a microchannel plate (MCP) with a phosphor screen and imaged with an OPAL-4000 CCD camera, providing a 2D image of continuous spectra of all the ions within the beam. The Thomson Parabola is calibrated in energy using a set of aluminum filter of different thickness, resulting in a minimum resolvable proton energy of $-0.6$ MeV, which is constrained by the size of the MCP. The raw image in Fig. 1 shows traces corresponding to copper, carbon, oxygen ions and protons. These accelerated ions originate from atoms located on the wire surface, including common contaminants such as hydrocarbons or water vapor. The majority of ions in the beam consist of protons, attributed to the TNSA sheath field's preferential acceleration of low charge-to-mass ratio ions. The resulting proton spectrum is fitted as

$$\frac{dN}{d\mathcal{E}} = \frac{c_s n_e t_{acc} S_{sheath}}{\sqrt{2\mathcal{E}T_h}} \exp(-\sqrt{2\mathcal{E}/T_h}), \qquad (3)$$

with $\mathcal{E}$ the particle energy, $t_{acc}$ the acceleration time, $c_s$ the sound speed and $S_{sheath}$ the electron sheath surface[43]. The slope temperature $T_h$ of the hot electrons on the rear surface can be inferred by fitting the proton spectra according to Eq. (3), as seen in Fig. 5. We obtain an average hot electron temperature of $400 \pm 30$ and $700 \pm 40$ keV with

standard deviation of 200 and 260 keV for the aluminum and copper targets, respectively. The variability in slope electron temperature can be attributed mainly to the narrow acceptance angle and distance of the TPS to the wire targets.

The hot electron temperature extracted from the ion spectra can then be compared to previously established collisionless laser-heating models[39,40]. For relativistic, short-pulse, high-contrast laser-solid interactions, the hot electron temperature is primarily determined by the normalized laser vector potential, $a_0 \approx 2.5$[60,61]. In this case, both the expressions found in Eq. (2) of Wilks et al.[39] and Eq. (10) of Haines et al.[40] give comparable hot electron temperatures, in the range of $T_h = 500-900$ keV, consistent with the experimentally obtained values.

## X-ray image processing

Due to defects and absorption effects in the beryllium CRL stacks, background removal techniques were used to reveal the density distribution inside the wire. Since the X-ray beam profile possesses a jitter in both pointing and intensity, the shot post-processing procedure relied on $\approx$200 images for both the X-ray background and the X-ray imaged cold (reference) wire target. The X-ray background is then removed on all shots, delivering a transmission map of the wire. An additional normalization of the shot image with a cold reference wire image $I_{shot}/I_{ref}$ is very effective in revealing changes in transmission induced by the density perturbations inside the target.

## Filament wavelength measurements

Since the 2D X-ray images obtained from the MXI are a projection of a 3D forest of filaments evolving within the target, it is necessary to ensure that the filament wavelength is conserved through its projection and imaging system. This is shown in Fig. 6. Approximating the filament density distribution with a sinusoidal signal, the wavelength conservation through the 3D structure can be explained via the linearity of the Fourier transform, meaning that a sum of multiple sinusoidal signals conserve the Fourier components of the individual signals. Since the accumulation of both absorption and phase contrast effects along the X-ray propagation axis can be described as a sum of the effects caused by individual sinusoidal filament rows, the wavelength components of the filament forest is conserved by the 2D projection.

Verifying the conservation of the filament wavelength through the imaging system requires modeling the imaging system synthetically to understand its effects on the obtained images. The phase effects are produced by the chromaticity of the beryllium CRLs, leading to a range of foci[45], which produce an effective propagation distance $d$ between the target and the imaging plane. This effective propagation distance is determined experimentally from undriven cold wire reference images. Analytically, we model the MXI as a perfect thin lens imaging system with a focusing error $\epsilon = 1/f - 1/d_o - 1/d_i$, where $f$, $d_o$, and $d_i$ are the focal length, object distance, and imaging distance.

Any material then modulates an incoming plane wave such that the function $f(x, y)$, describing the target plane before propagation, can be written as

$$f(x,y) = \underbrace{\exp\left[i\frac{2\pi}{\lambda}T(x,y)\right]}_{\text{In vacuum plane wave}}$$
$$\underbrace{\underbrace{\exp\left[-i\frac{2\pi}{\lambda}\delta T(x,y)\right]}_{\text{Phase change}} \underbrace{\exp\left[-\frac{2\pi}{\lambda}\beta T(x,y)\right]}_{\text{Absorption factor}}}_{\text{Transmission function}} \qquad (4)$$

where $T(x, y)$ is the two-dimensional thickness function of the object that the X-rays of wavelength $\lambda$ interact with, and $\delta$, $\beta$ are the real and imaginary part of the refractive index.

After passing through the target, the modulated plane wave in the input plane propagates in free space for a propagation distance

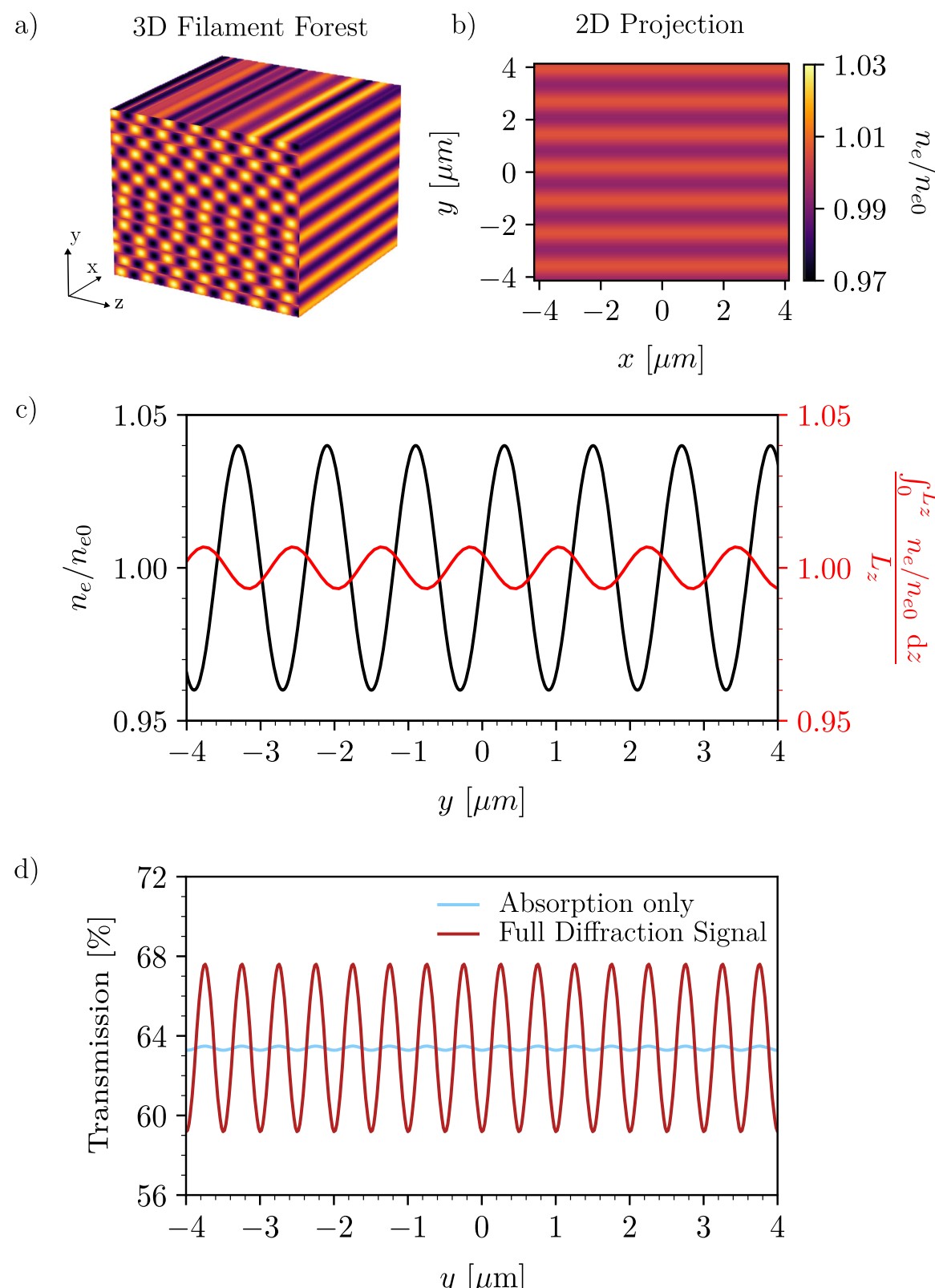

**Fig. 6 | Filament wavelength conservation through the X-ray imaging system.** X-rays propagating along z through a forest of filaments of length $L_z$ (**a**) conserve the filament wavelength in its 2D projection (**b**). This is further demonstrated by taking a 1D lineout of the projected 2D density distribution and comparing it to the input signal for the 3D filament forest (**c**). Additionally, the filament wavelength is conserved in the phase dominated diffraction signal (**d**) obtained from synthetic 2D X-ray images.

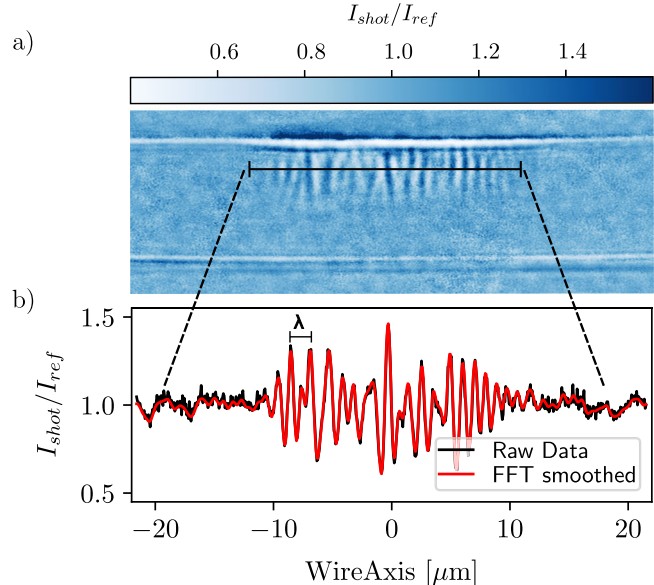

**Fig. 7 | Measuring the filament wavelength. a** X-ray image of the short-pulse laser interaction with a ⌀ 10 μm copper wire target. **b** Lineout of the filament structure measured at a penetration depth of 2.5 μm from the edge of wire target, showing a clear oscillatory structure with a distribution of different wavelengths and amplitudes (black). The lineout is smoothed via Fourier filters to aid the analysis (red). The peak-to-peak distance in the lineouts is used to extract the filament wavelength $\lambda$.

$d = d' - d_0 \approx -\epsilon d_0^2$ to an output plane $g(x, y)$. The output function can then be described as the product of the transfer function $\Psi(\nu_x, \nu_y) = \exp\left[i\pi\epsilon d_0^2\lambda(\nu_x^2 + \nu_y^2)\right]$ with the Fourier transform of the input function in Fourier space[62]

$$g(x,y) = \Psi_0 \int\int_{-\infty}^{\infty} F(\nu_x, \nu_y) \exp\left[-i\pi\lambda d(\nu_x^2 + \nu_y^2)\right]$$
$$\exp\left[-i2\pi(\nu_x x + \nu_y y)\right] \mathrm{d}\nu_x \mathrm{d}\nu_y, \tag{5}$$

with $\Psi_0 = \exp(id2\pi/\lambda)$. Since the propagated wave attained from $g(x, y)$ is subsequently imaged on the detector, a synthetic X-ray image can be calculated via $I_{\text{synthetic}}(x, y) = |g(x, y)|^2$. We can extract the full Fresnel diffraction pattern on our detector produced by a thickness mask representing the targets in the experiment. To emulate the limitation in the resolution of the imaging system, a Gaussian filter is applied to the synthetic images to artificially limit the resolution to ≈200 nm.

The synthetic X-ray images are used to systematically analyze the response of the imaging system to sinusoidal density perturbations. Due to the strong sensitivity to density gradients the imaging system appears to distort the signal in the case of very strong density perturbations concentrated in small filaments. However, even in such scenarios the overall wavelength, meaning the distance between same polarity filaments, remains conserved.

The filament wavelength is extracted via 1D lineouts taken at various distances from the target front surface. A Fourier filter is applied to remove the high and low frequency components of the signal as seen in Fig. 7. The resulting signal exhibits a range of different wavelengths and amplitudes clearly distinguishable from the background. Each maximum and minimum, corresponding to filaments of opposite polarity, is individually fitted by a Gaussian distribution function. To account for the susceptibility of the imaging system to extreme electron density gradients, filaments are organized in pairs of opposite polarity. The distance between each filament pair is then used

to extract a distribution of filament wavelengths for each shot and distance from the front surface.

## Filament merging model

The merging of filaments after the saturation of the instability can be modeled using a simplified approach, following ref. 47. The main steps of the derivation are reproduced here for completeness. We consider a distribution of straight, internally uniform filaments with identical current, polarity, mass density and radius, separated by a distance $D_0 = \lambda_0/2$, and interacting pair-wise via the attractive force per unit length, $dF/dL = -B_0 I_0/c$. Here $B_0 = 2I_0/(cr)$ is the magnetic field associated to a filament and $r$ is the distance to the axis. Interactions involving more than two filaments or filaments of opposite polarities are neglected. The equation of motion becomes

$$\frac{\mathrm{d}^2 x}{\mathrm{d}t^2} = -\frac{2I_0^2}{c^2\mu_0 x}, \tag{6}$$

where $\mu_0$ is the lineic mass. In the asymmetric regime, the mass density carried during the merging process must be dominated by that of the ions $mn \approx m_i n_i$ to maintain quasi-neutrality even if the current is carried by the electrons, such that $\mu_0 = nm_i(\pi D_0^2/4)$. The initial merging time, that is, the time for two filaments to cross the distance $D_0$, may be evaluated analytically from Eq. (6) as

$$\tau_{m0} = \frac{\sqrt{\pi\mu_0}cD_0}{2I_0 \,\mathrm{erfc}(\sqrt{\log 2})} \approx 0.39(m_i n)^{1/2}\frac{\lambda_0}{B_0}. \tag{7}$$

The pair-wise merging process occurs in a self-similar way, since each step of filaments merging gives rise to identical starting conditions but with rescaled parameters. After the $k$th merging step, the current and lineic mass of the filaments evolve as $I_k = 2^k I_0$ and $\mu_k = 2^k\mu_0$, and the separation distance from their neighbors increases as $D_k = 2^{k/2}D_0$ such that $\lambda_k = 2^{k/2}\lambda_0$. The time required to reach the $k$th step is $t = \sum_{k'=0}^{k} \tau_{mk'}$, hence $k = t/\tau_{m0}$ since the merging time is independent of $k$. The resulting wavelength evolution may then be calculated as

$$\lambda(t) = \lambda_0 2^{t/(2\tau_{m0})} \tag{8}$$

and can be used to infer the magnetic field if the merging time and initial wavelength are known.

By fitting this model to the experimental measurements shown in Fig. 3, the merging time is $\tau_m = 11.0 \pm 0.4$ ps for copper and $8.53 \pm 0.25$ ps for aluminum. Using Eq. (2) the inferred magnetic field is $B_0 = 10.64 \pm 0.13$ MG and $B_0 = 7.8 \pm 0.1$ MG for copper and aluminum, respectively. Here, the uncertainties are derived from standard error propagation based on the determination of the wavelength, the noise level of the background, as well as fitting errors and do not account for the simplifying assumptions of the model.

## Particle-in-cell simulations

The two-dimensional simulations of the laser-target interaction have been performed for the ⌀ 10 μm copper wire with the relativistic and massively parallel PIC code OSIRIS 4.0[53]. The laser irradiation of the solid density target and subsequent plasma evolution are described fully kinetically, using a realistic ion-to-electron mass ratio and realistic plasma density $n_i = 8.49 \times 10^{22}$ cm$^{-3}$. The copper ions are considered to be ionized 7 times, based on the results of a 0D three-temperatures model of the laser-target interaction[50]. The PIC simulations include Coulomb collisions between all populations using a relativistic Monte-Carlo operator[63].

The 2D simulation domain models a slice of the wire with a box size $L_x$ (wire radius direction) = $L_y$ (wire axis direction) = 40 μm and spatial resolution of $6.9 \times 10^{-3}$ μm. The boundary conditions for both particles and fields are open along the x direction and periodic along

the y direction. The interaction is followed for several picoseconds with a time step $\Delta t = 0.011$ fs to ensure proper calculation of the collisions and avoid numerical heating. The P-polarized laser pulse with a wavelength of 0.8 μm and normalized laser potential $a_0 = 3$ is focused on the wire front surface. It follows a Gaussian temporal profile in intensity of 150 fs FWHM and a Gaussian spatial profile with 6 μm FWHM transversely to the target normal. The pre-plasma is modeled assuming an exponentially decreasing density profile with a scale length of 0.07 μm from solid density down to $n_e/n_{crit} = 0.1$. We used 400 macroparticles per cell per species, with cubic particle shape for improved numerical accuracy when depositing the current on the grid. We have tested the convergence of the results by varying the spatial resolution ($0.5–1 c/\omega_{pe}$), the time step ($0.49 − 0.71 \omega_{pe}^{-1}$), the number of particles per cell per species (64–400), the box size ($L_x = 30–40$ μm, $L_y = 40–80$ μm) and the pre-plasma scale length (0.03–0.12 μm). We also performed a 2D simulation with a circular target (top-view) and found that it did not significantly affect the fast electron properties and the filamentary structures that develop in the dense plasma.

## Dispersion relation of the current filamentation instability

The growth rate and most unstable wavelength of the current filamentation instability can be calculated from the dispersion relation derived for a system of two drifting Maxwell–Boltzmann electron distributions:

$$f_{0s}(\mathbf{v}) = \frac{n_{0s}}{2\pi v_{ths}^2} \exp\left[ -\frac{(v_x - v_{ds})^2 + v_y^2}{2 v_{ths}^2} \right], \tag{9}$$

where the index $s = h, c$ corresponds to either the hot and return current populations and $v_{ths} = (T_s/m_s)^{1/2}$ is the thermal velocity. We consider the electron counter-streaming velocity $v_{ds}$ to be in the $\hat{\mathbf{e}}_x$ direction and, without loss of generality, the wavenumber of the unstable filamentation mode to be $k_y$. The choice of non-relativistic distributions is justified by the dominant role of the return current, which is non-relativistic, and the relative insensitivity to the hot electron temperature in the conditions of the experiments as we will discuss below. The return-current electron population provides current neutralization and a cold non-drifting background ion distribution provides charge neutralization $Z n_{0i} = n_{0e}$.

**Collisionless filamentation.** Following the standard linearization of the Vlasov-Maxwell equations, considering the contributions of both electromagnetic ($\mathbf{E}_{EM} = E_x \hat{\mathbf{e}}_x$) and electrostatic ($\mathbf{E}_{ES} = E_y \hat{\mathbf{e}}_y$) fields, and perturbations of the type $\propto e^{i(k_y y - \omega t)}$ the collisionless dispersion relation is given by[25,26]

$$\det\left\{ \begin{pmatrix} \omega^2 - \omega_{pi}^2 - c^2 k_y^2 & 0 \\ 0 & \omega^2 - \omega_{pi}^2 \end{pmatrix} + \right.$$
$$\left. \sum_s \omega_{ps}^2 \begin{pmatrix} -\left(\zeta_s^2 + \frac{1}{2}\right)\mathcal{Z}_s' & \zeta_s \mathcal{Z}_s + \frac{\zeta_s}{2}\mathcal{Z}_s'' \\ \zeta_s \mathcal{Z}_s + \frac{\zeta_s}{2}\mathcal{Z}_s'' & -\frac{3}{2}\mathcal{Z}_s' - \frac{1}{4}\mathcal{Z}_s''' \end{pmatrix} \right\} = 0, \tag{10}$$

where

$$\mathcal{Z}_s \equiv \mathcal{Z}(\xi_s) = \frac{1}{\sqrt{\pi}} \int_{-\infty}^{\infty} dt \frac{e^{-t^2}}{t - \xi_s} \tag{11}$$

is the plasma dispersion function with $\xi_s = \omega/(\sqrt{2} k_y v_{ths})$, $\zeta_s = v_{ds}/(\sqrt{2} v_{ths})$ and the number of primes in the $\mathcal{Z}$ functions denotes the number of differentiations. The two diagonal terms are responsible for the electromagnetic and electrostatic modes, whereas the off-diagonal terms give rise to the coupling between both modes.

Reduced expressions for the fastest growing mode can be obtained in the case of purely electromagnetic perturbations, i.e.,

neglecting the coupling to electrostatic modes. Considering $\omega = i\Gamma$ with $\Gamma$ the growth rate of the filamentation instability, and taking the relevant limits for the conditions of the experiment and simulations $\alpha = n_h/n_0 \ll 1$, $v_{dh} \ll v_{thh}$ and $v_{dc} \ll v_{thc}$ (corresponding to the kinetic limit), the dispersion relation becomes insensitive to the hot electron temperature and is given by

$$\Gamma^2 + (1 - \alpha)\sqrt{\frac{\pi}{2}} \frac{\Gamma}{k\beta_{thc}} - \frac{\alpha^2}{1 - \alpha}\frac{\beta_{dh}^2}{\beta_{thc}^2} + k^2 = 0. \tag{12}$$

In Eq. (12) and hereafter the growth rate $\Gamma$ is normalized by the electron plasma frequency $\omega_{pe} = (4\pi n_0 e^2/m_e)^{1/2}$, the wavenumber $k$ is normalized by $\omega_{pe}/c$ and $\beta = v/c$ is the normalized velocity. The wavenumber subscript has been omitted to simplify the notation. To leading order in $k^2 \beta_{thc}^2 \ll 1$ the collisionless dispersion relation becomes simply

$$\Gamma = \sqrt{\frac{2}{\pi} \frac{k\beta_{thc}}{1 - \alpha} \left( \frac{\alpha^2}{1 - \alpha}\frac{\beta_{dh}^2}{\beta_{thc}^2} - k^2 \right)}. \tag{13}$$

The maximum growth rate is

$$\Gamma_{max} = \frac{2}{3}\sqrt{\frac{2}{3\pi}} \frac{\alpha^3}{(1 - \alpha)^{5/2}} \frac{\beta_{dh}^3}{\beta_{thc}^2} \tag{14}$$

and corresponds to the most unstable wavenumber

$$k_{max} = \frac{\alpha}{\sqrt{3(1 - \alpha)}} \frac{\beta_{dh}}{\beta_{thc}}. \tag{15}$$

In this regime, the properties of the return current control the growth of the instability with a growth rate that is much smaller than the plasma frequency.

By numerically solving the full collisionless dispersion relation of Eq. (10), we find that space-charge effects play an important role in this regime further decreasing the growth rate. The results for the conditions informed by PIC simulations, namely $n_{0e} = 5.8 \times 10^{23}$ cm$^{-3}$, $\alpha = 0.02$, $\beta_{dh} = 0.3$, $\beta_{thc} = 0.45$, $T_e = 100$ eV, $Z = 7$ and $m_i/m_e = 1.158 \times 10^5$, are shown in Fig. 4d.

**Resistive filamentation.** The importance of collisional effects in the dense plasma regions of solid-density target or ICF targets have led to significant efforts in studying the impact of collisions on the filamentation instability. Given the much lower density of the hot electrons with respect to the background in these scenarios, collisions affect primarily the non-relativistic return current population.

Most previous work has explored the impact of collisions by including a Bhatnagar–Gross–Krook (BGK) collisional term $v(f - f_0)$ on the right-hand side of the Vlasov equation[26,27]. These works found that collisions could either enhance or mitigate the filamentation instability, depending on the regime. The simplicity of the BGK operator makes it appealing, but it is limited to the weakly collisional regime, where the collision time is significantly larger than the plasma collective times, on top of additional drawbacks related for example to the non-relaxation to a Maxwell–Boltzmann distribution and the non-conservation of particle number.

In the case of the current experimental and simulation conditions, the collision frequency is very large. For example, for a copper plasma with $n_{0e} = 5.8 \times 10^{23}$ cm$^{-3}$, $Z = 7$ and $T_e = 100$ eV the electron collision frequency is $v_e \approx 0.16 \omega_{pe}$[63], which is much larger than the growth rate of the collisionless filamentation instability ($\Gamma \approx 4 \times 10^{-5} \omega_{pe}$).

In order to analyse the role of collisional effects in this regime, we consider a resistive model of the filamentation instability containing the same two counter-streaming electron distributions given by Eq. (9) and a charge neutralizing ion population, but where the electric field is now given by a simple Ohm's law $\mathbf{E} = \eta \mathbf{J}_c$ with $\eta$ the electrical resistivity.

Following the linearization of the Vlasov-Maxwell equations, the resulting dispersion relation for electromagnetic fluctuations is given by[34]

$$\Gamma^2 + (1-\alpha)\frac{\Gamma}{\nu_e} - \alpha\left\{\frac{\beta_{dh}^2}{\beta_{thh}^2} - \left(1 + \frac{\beta_{dh}^2}{\beta_{thh}^2}\right)\right.$$
$$\left. \sqrt{\frac{\pi}{2}}\frac{\Gamma}{k\beta_{thh}}\exp\left(\frac{\Gamma^2}{2k^2\beta_{thh}^2}\right)\left[1 - \mathrm{erf}\left(\frac{\Gamma}{\sqrt{2}k\beta_{thh}}\right)\right]\right\}$$
$$+ \frac{Zm_e}{m_i} + k^2 = 0.$$

(16)

To leading order in $\Gamma/(k\beta_{thh}) \ll 1$ and $\nu_e^2\alpha\beta_{dh}^2/\beta_{thh}^2 \ll 1$ the growth rate is

$$\Gamma = \frac{\alpha\frac{\beta_{dh}^2}{\beta_{thh}^2} - k^2}{\frac{1-\alpha}{\nu_e} + \sqrt{\frac{\pi}{2}}\frac{\alpha}{k\beta_{thh}}\left(1 + \frac{\beta_{dh}^2}{\beta_{thh}^2}\right)}.$$

(17)

The comparison of the full resistive (Eq. (16)) and collisionless (Eq. (10)) dispersion relations is shown in Fig. 4d. We find that in the regime of interest for the experiments and simulations, the main role of collisional effects is to shift the most unstable wavelength towards larger scales.

## Data availability

The source data that support the findings of this study are part of ongoing analyses for future publications and can be made available from corresponding author C.S. upon request.

## Code availability

The PIC code OSIRIS used in this study can be obtained from the OSIRIS Consortium, consisting of UCLA and IST (Portugal).

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

## Acknowledgements

We would like to acknowledge the MEC team for the development of the MXI diagnostic and experimental support, and E.P. Alves and F. Seiboth for their support in modeling the X-ray imaging system. This work was performed under the auspices of the U.S. Department of Energy (DOE) by SLAC National Accelerator Laboratory under FWP 100182 and by SLAC Laboratory Directed Research and Development. Use of the Linac Coherent Light Source (LCLS) SLAC National Accelerator Laboratory is supported by the U.S. Department of Energy Office of Science (SC), Basic Energy Science under Contract No. DE-AC02-76SF00515. Matter in Extreme Conditions research at LCLS is supported by SC, Fusion Energy Science, FWP 100106 under Contract No. DE-AC02-76SF00515. A.M. acknowledges the support from the US DOE Early Career Research Program under FWP 100331. G.D.G. acknowledges support from the DOE NNSA SSGF program under DE-NA0003960 and in part, NSF Grant PHY-2308860 for NSF funding. F.F. acknowledges support from the European Research Council (ERC-2021-CoG Grant XPACE No. 101045172). The simulations were run on Perlmutter (NERSC) through an ALCC award.

## Author contributions

C.S., M.G., and F.F. conceived and led this project. C.S., M.G., S.A., C.B.C., E.C., G.D., S.F., G.D.G., S.G., D.K., M.R., U.S., F.T., M.V., K.Z., and S.H.G. designed and executed the experiment. C.S., M.G., F.T., and G.D.G. analyzed the experimental data. A.M. and F.F. performed the simulations and theoretical analysis. C.S., A.M., F.F., and M.G. interpreted the experimental results. The paper was written by C.S., A.M., F.F., and M.G. with contributions from all the authors.

## Competing interests

The authors declare no competing interests.
