## [Transparent Peer Review file · Nature Communications]

Time-resolved X-ray imaging of the current filamentation instability in solid-density plasmas

Corresponding Author: Dr Christopher Schoenwaelder

Version 0:

Reviewer comments:

Reviewer #1

(Remarks to the Author)

In the manuscript "Time-resolved X-ray imaging of the current filamentation instability in solid density plasmas" authors present their recent experiment on the current filamentation in the laser plasma, where the X-ray free electron laser (XFEL) was used for probing the developing density modulations. The study is relevant for such fields of research as astrophysics, inertial confinement fusion, laser plasma ion acceleration, etc. Direct experimental characterization of the current filamentation instability in the solid density laser plasmas is very challenging as it requires penetration of such media by the probe. In the earlier studies, the laser-plasma accelerated protons were considered to probe the fields within plasma, as they have sufficiently short duration and proper mass-to-charge ratio for this task. The original feature of the present study and is the use of XFEL pulses along with the multi-TW laser that can drastically improve the temporal and spatial resolution in measuring directly the plasma density. Currently, the access to such joined XFEL and laser-plasma experiments is very limited, which puts this work in rather a unique position.

The experiments had the single-angle pump-probe setup with a wire target, and the measurements included the X-ray phase-contrast plasma imaging, and the ion spectra measurements with the Thomson parabola. The plasma density filaments were observed over the first tens of picoseconds after the laser, and were analyzed in terms of their amplitude and spatial frequency. Comparing these observables with the simple theoretical and 2D PIC simulations, allowed estimating the electron plasma temperature and magnetic fields, as well as determining the role of such processes as ion motion and collisions in the instability development and termination.

The paper is well written and explains the material comprehensively. The presented experimental results are new and they demonstrate how XFEL pulses can be used to probe fine structures in solid plasmas. In my opinion, the paper may be published in Nature Communications, but before recommending it I would like to suggest following remarks:

- the estimates of electron temperature and magnetic field are provided with high numerical accuracy (e.g. $B_0=10.64\pm 0.13$) and rather the modest errorbars supposedly obtained from fitting with theoretical models (e.g. Eqs (1,2), PIC modeling). To me it feels not very rigorous, as all these models are based on strong simplifications (reduced geometry, pre-ionized plasma, identical filament currents etc), so they cannot physically provide such an accuracy, and this may be misleading as it gives the impression that this method can indeed be used to measure magnetic field inside the plasma with 1% precision. The validity of the estimates should be discussed thoroughly and without any implicit exaggerations.

- line 122: authors mention that electron temperature deduced from ion spectra agree with heating models. it would be instructive provide the used expressions for the temperature and the conditions under which it was applied;

- lines 177 and 186 : authors provide the estimate of electron current velocity of $\sim 0.5c$ but I've failed to find where this number is coming from -- intuitively it makes sense, but it should be defined by the interaction of the laser with the pre-plasma and plasma skin-layer and may vary depending on the actual heating mechanism.

After authors address these comments I would be able to recommend this manuscript for the publication.

Reviewer #2

(Remarks to the Author)

The paper "Time-resolved X-ray imaging of the current filamentation instability in solid density plasmas" by Schoenwaelder et al. describes the use of high resolution x-ray imaging to diagnose the current filamentation instability in solid density copper plasmas.

However I am not convinced that the authors' explanation for these measurements is justified.

The spatial resolution of the diagnostic is impressive, and it shows filamentary structures apparently expanding into the 5-micron wire. The filaments must be electron and ion density modulations to be measurable – and since they persist over > 10 of picoseconds. (100 x greater than the laser pulse duration!) However, it seems strange that hot electron populations are trapped in a filament for time duration of this length. There have been many previous measurements of the duration of x-ray pulses from such experiments (K-alpha etc) and the time scale is much less. Other work has shown the hot electrons refluxing throughout the plasma and propagating very far from the interaction region.

The experimental work is accompanied by PIC simulations which show some similar behavior on shorter timescales, shorter spatial scales and in 2D while also using a very small initial density scalelength.

The data is very nice but there has also been a lot of work previously on this instability in HED plasmas– not all of which has been referenced.

1) Experimental Study of Current Filamentation Instability

B. Allen et al., Phys. Rev. Lett. 109, 185007 (2012)

2) Observations of the filamentation of high-intensity laser-produced electron beams

M. S. Wei et al., Phys. Rev. E 70, 056412 (2004)

3) Weibel-Induced Filamentation during an Ultrafast Laser-Driven Plasma Expansion

K. Quinn et al., Phys. Rev. Lett. 108, 135001 (2012)

4) Dynamics of the Electromagnetic Fields Induced by Fast Electron Propagation in Near-Solid-Density Media L. Romagnani et al. Phys. Rev. Lett. 122, 025001 (2019).

5) Current filamentation instability in laser wakefield accelerators

C M Huntington et al., Phys Rev Lett., 106 :105001 (2011)

Questions:

1) Why are the simulations not 3D? It seems that the cylindrical geometry of the wire will play a significant effect on the physics – especially since the wire diameter is about the same as the focal spot diameter.

2) Why do the filaments only extend to the center of the wire? This would be expected if the density modulations observed are due to plasma structures on the outside surface of the wire (i.e. the area that has been irradiated). How do the authors know that the density modulations are not due to such hydrodynamics structure (perhaps induced by pre-pulses) on the irradiated surface of the wire.

3) The experimental electron temperature is derived from the slope of the proton spectrum. This is problematic since the proton spectra is highly dependent on the laser pulse duration (total pulse energy) rather than the peak intensity. There are many review papers about this.

4) What is the laser contrast and the laser and the expected (or measured) density scalelength?

Reviewer #3

(Remarks to the Author)

The Ms "Time-resolved X-ray imaging of the current filamentation instability in solid density plasmas" reports about imaging laser-matter interaction with very high time resolution using the combination of a FEL and probing laser-matter interaction. The topic of radiography is always an interesting topic within the community as far as it can unlock some bottlenecks within this research area. As such, the paper could be interesting for the community. However, probing of matter using X-rays has already been done in the past, such as using laser-accelerated protons to resolve quick electromagnetic phenomena (e.g. see the PRL by Romagnani 2005). For the moment it is therefore not completely clear, what superiority the quite complex setup used by the author has compared to what has been used in the past and what additional information can be retrieved with this complex setup in addition to what other methods would be able to detect. I think that authors should emphasize this in their Ms such as to convey a clearer message and show superiority of their (quite complex) setup compared to "easier" solutions using only X-rays or laser-accelerated protons.

More in detail:

Abstract: authors write "Our findings indicate that magnetic fields on the order of 10 megagauss are produced, which will help confine high-energy electrons within the filaments with important implications for the transport of energetic particles in plasmas."

What about electric fields? Are they not of importance? What implications can this have? What applications have authors in mind that could have an impact on this?

Page 3: the resolution seems to be 50 fs and <200 nm. Again, what would be the resolution using laser-accelerated protons (that can probe at ps time frames) or laser-accelerated X-rays, is a resolution of 200 nm required?

Page 5: it is not completely clear what "The temporal evolution of the filament wavelength" is indicating? How has this

information been retrieved from Fig. 2 ? Maybe a small explanation (arrows etc) could help the reader understand the underlying calculations.

Page 7: the statement "These experimental results reveal a critical importance of electrostatic effects and ion motion in the dynamics of the asymmetric current filamentation instability" - how is this importance revealed ? The ion motion seems to be a new information, how is that information retrieved ?

Page 8 (and theory): it would be interesting to compare the PIC results to the experimental results. Authors state that there is a good agreement in the text, but finally there is no figure comparing the results, which would strengthen their statement, even visually.

Page 9: Author mention: "Our results reveal the combined importance of space-charge and resistive effects on the growth and nonlinear dynamics of the current filamentation instability." - What does this mean ? Are the two, space-charge and resistive effects, individually important, or only its combination ? What is the impact on the filamentation individually.

In general, the main message of the paper is not fully clear: Initially it seems that the main message is about a new imaging diagnostics ("Time-resolved X-ray imaging"). Then it evolves to the importance of magnetic fields in plasma and finally to filamentation issues, space-charge and resistive effects. I would encourage the authors to find a guideline ("fil conducteur") related to what they want to say within the Ms and build the content/results/discussion accordingly.

Version 1:

Reviewer comments:

Reviewer #1

(Remarks to the Author)

I have read carefully the revised manuscript "Time-resolved X-ray imaging of the current filamentation instability in solid-density plasmas" and the response of author to the referees' remarks. I have found my critics well addressed, as well as I've appreciated the detailed and clear responses to the rigorous remarks of the other referees. In the current form I can recommend the manuscript for publication as an article in Nature Communications.

Reviewer #2

(Remarks to the Author)

The authors of "Time-resolved X-ray imaging of the current filamentation instability in solid-density plasmas" have responded to my previous review and have added further explanations and references. However on the whole I do not believe that this paper is suitable for Nature Communications. The diagnostic is very impressive, however the Weibel instability has been explored in many other previous papers.

In addition there are still many aspects about this explanation that are not convincing including the density of ion perturbations probed, the duration of the filamentary structures, the length of the filaments and the 2D numerical modeling which lasts only 2 psec. The observed structures are much more likely to be due to hydrodynamics on the wire surface or due to the electrothermal instability as in previous observations.

Reviewer #3

(Remarks to the Author)

The authors made significant efforts for clarifying the concerns of the reviewers and the paper is now much more robust and understandable. Well done.

From my side, the logical sequence is still difficult to follow: the paper being very long (10 page), a small introductory sentence guiding the reader in the "Result" section would help, but this is up to the authors.

We thank the Reviewers for carefully reading our manuscript and for their constructive comments, which have helped us improve our manuscript. Below, we reproduce the Reviewers report (in blue), provide a detailed point-by-point response to each comment (in black) and indicate the corresponding changes in the manuscript. All changes made in the manuscript are color-highlighted in the revised version.

We are confident these changes have helped clarify the importance and novelty of our results on the characterization of a fundamental plasma instability in high-energy-density conditions, which has long been a significant experimental challenge and is relevant to a wide range of systems and applications, from high-energy astrophysics to the energy transport in inertial fusion plasmas to the development of laser-driven compact secondary sources. We hope that this work can now be accepted for publication in *Nature Communications*.

Reviewer #1:

In the manuscript "Time-resolved X-ray imaging of the current filamentation instability in solid density plasmas" authors present their recent experiment on the current filamentation in the laser plasma, where the X-ray free electron laser (XFEL) was used for probing the developing density modulations. The study is relevant for such fields of research as astrophysics, inertial confinement fusion, laser plasma ion acceleration, etc. Direct experimental characterization of the current filamentation instability in the solid density laser plasmas is very challenging as it requires penetration of such media by the probe. In the earlier studies, the laser-plasma accelerated protons were considered to the probe the fields within plasma, as they have sufficiently short duration and proper mass-to-charge ratio for this task. The original feature of the present study and is the use of XFEL pulses along with the multi-TW laser that can drastically improve the temporal and spatial resolution in measuring directly the plasma density. Currently, the access to such joined XFEL and laser-plasma experiments is very limited, which puts this work in rather a unique position.

The experiments had the single-angle pump-probe setup with a wire target, and the measurements included the X-ray phase-contrast plasma imaging, and the ion spectra measurements with the Thomson parabola. The plasma density filaments were observed over the first tens of picoseconds after the laser, and were analyzed in terms of their amplitude and spatial frequency. Comparing these observables with the simple theoretical and 2D PIC simulations, allowed estimating the electron plasma temperature and magnetic fields, as well as determining the role of such processes as ion motion and collisions in the instability development and termination.

The paper is well written and explains the material comprehensively. The presented experimental results are new and they demonstrate how XFEL pulses can be used to probe fine structures in solid plasmas. In my opinion, the paper may be published in Nature Communications, but before recommending it I would like to suggest following remarks:

We would like to thank the Reviewer for highlighting the novelty and broad importance of our study as well as the unique characterization of the filamentation instability enabled by it, which

has long been a significant experimental challenge. Below, we address, point-by-point, the Reviewer's remarks which have helped us improve our work.

The estimates of electron temperature and magnetic field are provided with high numerical accuracy (e.g. $B_0=10.64\pm 0.13$) and rather the modest errorbars supposedly obtained from fitting with theoretical models (e.g. Eqs (1,2), PIC modeling). To me it feels not very rigorous, as all these models are based on strong simplifications (reduced geometry, pre-ionized plasma, identical filament currents etc), so they cannot physically provide such an accuracy, and this may be misleading as it give the impression that this method can indeed be used to measure magnetic field inside the plasma with 1% precision. The validity of the estimates should be discussed thoroughly and without any implicit exaggerations.

The values for the precision of the inferred magnetic field are obtained from standard error propagation based on the determination of the wavelength, the noise level in the X-ray images, as well as fitting errors of the theoretical model to the experimental data. The Reviewer is correct that they do not account for the underlying simplifying assumptions of the model and we agree that this should be clarified.

We have revised the main text to give the approximate values of the magnetic field and make clear how these are obtained from fitting to the experimental measurements (lines 189 - 193 in revised manuscript). We also added a more detailed discussion to the methods (section: Filament merging model, lines 456-462 in revised manuscript), where we include the error bars from the procedure above, and explain that they do not account for the simplifying assumptions of the underlying model.

line 122: authors mention that electron temperature deduced from ion spectra agree with heating models. it would be instructive provide the used expressions for the temperature and the conditions under which it was applied;

The expressions used correspond to Eq. (10) and Eq. (11) of [Haines, M. G. et al., Physical Review Letters 102, 045008 (2009)], with Eq. (11) describing the pondermotive scaling as found by [Wilks, S. C. et al., Physical Review Letters 69, 1383–1386 (1992)]. These models work under the assumption of relativistic, short-pulse, laser-solid interactions, which is the relevant regime in our MEC experiments with a high-contrast laser and a laser vector potential $a_0 = 2.5$ [Malka, G. & Miquel, Physical Review letters 77, 75-78 (1996); Rusby et al. Physics of Plasmas 31, 040503 (2024)]. In this regime T_{hot} is primarily determined by the normalized laser vector potential. Both models give comparable temperatures in the range $T_{\text{hot}} = 500 - 900$ keV, consistent with the values inferred from the measured ion spectrum.

We have modified the sentence addressing the heating models in the main text to state that these models apply for high-contrast laser-plasma interactions (lines 127 - 130 in revised manuscript). We have also added a more detailed discussion in the methods section ("Ion energy spectrum and hot electron temperature", lines 364 - 370 in revised manuscript), that

includes how the hot electron temperature is extracted from the ion spectra, as well as the references to the hot electron temperature models and their underlying assumptions.

lines 177 and 186 : authors provide the estimate of electron current velocity of $0.5c$ but I've failed to find where this number is coming from -- intuitively it makes sense, but it should be defined by the interaction of the laser with the pre-plasma and plasma skin-layer and may vary depending on the actual heating mechanism.

Indeed, the details of the electron distribution function do depend on the laser and plasma properties. However, for relativistically intense lasers and solid-density (overdense) targets, the average energy of these electrons can be estimated based on the slope temperature of the laser-accelerated electron energy distribution, which scales primarily with laser intensity (or normalized vector potential) as discussed above. For the conditions of our experiments T_{hot} is a few hundred keV. Based on this, it is reasonable to estimate that a typical velocity of the fast electrons in the forward direction is a large fraction of c , and we have thus considered a value of $0.5c$. This is of course a simple estimate for the typical electron drift velocity and is in reasonable agreement with the average fast electron drift velocity measured in the simulations $v=0.3c$.

We have revised the text in the manuscript to make this clearer (lines 195 - 198 in revised manuscript).

Reviewer #2:

The paper "Time-resolved X-ray imaging of the current filamentation instability in solid density plasmas" by Schoenwaelder et al. describes the use on high resolution x-ray imaging to diagnose the current filamentation instability in solid density copper plasmas.

However I am not convinced that the authors' explanation for these measurements is justified.

The spatial resolution of the diagnostic is impressive, and it shows filamentary structures apparently expanding into the 5-micron wire. The filaments must be electron and ion density modulations to be measurable – and since they persist over > 10 of picoseconds. (100 x greater than the laser pulse duration!) However, it seems strange that hot electron populations are trapped in a filament for time duration of this length. There have been many previous measurements of the duration of x-ray pulses from such experiments (K-alpha etc) and the time scale is much less. Other work has shown the hot electrons refluxing throughout the plasma and propagating very far from the interaction region.

We thank the Reviewer for the careful consideration of our work and for recognizing the impressive nature of our measurements. We are happy to further clarify the importance of these measurements and their interpretation.

As the Reviewer mentions, it is clear that the observed filaments are due to electron and ion density modulations in the plasma and must persist for >10 ps, as discussed in the manuscript. This, in itself, is already a very important result and observation of our work. As we will discuss below, it is also clear that these modulations occur inside the dense plasma, thus constituting a very important and long-sought characterization of the density evolution of the filamentation instability in high-energy-density plasmas.

The Reviewer is correct that, on these time scales, the hot electrons can recirculate within the target, although this does not prevent them from driving a return current that is unstable to the current filamentation instability. In the absence of strong magnetic fields, if electrons move ballistically, an important fraction will leave the central (interaction) region along the wire axis due to their divergence, and the number of hot electrons in the interaction region will decrease over time, which can affect the long-time dynamics of the instability. We do not measure the fast electron dynamics and lifetime directly in the experiment. However, if strong magnetic fields are produced with magnitudes on the order of the Alfvén limit, as inferred from the experimental analysis and observed in the corresponding simulations, then, as we argue in the paper, this could significantly affect their transport along the axis of the wire, as these fields are perpendicular to it. For example, while the electrons will continue to recirculate in the target and some fraction can still escape from the central region, once such strong fields are established they would help confine the electrons and significantly limit their escape. In this case, one would expect their lifetime to be primarily determined by collisions within the plasma as they recirculate in the target. As indicated in the manuscript, the collisional time for 700 keV electrons is ~ 20 ps, consistent with the duration of the density modulations observed experimentally. At the same time, we note that it is the return current electrons that drive the instability and the relative importance of these two populations for the lifetime of the filaments is not yet well established, neither theoretically nor experimentally.

We agree that complementing our measurements with simultaneous K-alpha or X-ray Thomson scattering diagnostics would be very interesting and in fact we have just submitted a proposal to do that. However, we would like to mention a couple of points regarding the relation between K-alpha, X-ray emission, and the lifetime of fast electrons in these type of experiments:

1) We are not sure which references the Reviewer is referring to. It is correct that there have been publications indicating sub-picosecond K-alpha durations [e.g. Feurer et al. *Appl Phys B* 72, 15–20 (2001)]. However, there have also been several experiments using ~ 150 fs duration lasers indicating K-alpha emission for durations of > 10 ps (much longer than the laser) [e.g. Andiel et al. *Phys. Rev. E* 63, 026407 (2001); Bastiani-Ceccotti et al. *Appl Phys B* 78, 905–909 (2004); Shepherd et al. UCRL-TR-229579, 908084 (2007); Nagel et al. *Appl. Phys. Lett.* 110, 144102 (2017); Sawada et al. *Nature Communications* 15, 7528 (2024)];

2) If K-alpha emission originates primarily from fast electrons in the target, one could expect a shorter signal associated with the burst of fast electrons that are produced during the laser-pulse duration, followed by a longer duration signal associated with the refluxing of these electrons in the target and their collisional equilibration process. The relative strength of these

two emission phases will depend on the details of the interaction, the target size, and the generation of fields inside the plasma. Moreover, the ability to clearly detect and separate them will depend on the sensitivity of the diagnostics and on their temporal resolution;

3) To our knowledge, a clear characterization of these phases using K-alpha has not been established in these types of experiments and as such the connection between K-alpha diagnostics and the presence/properties of fast electrons needs to be more carefully understood.

We have revised the manuscript to indicate that the life time of the fast electrons in the target is not well established and that it would be interesting to have simultaneous X-ray Thomson scattering or K-alpha diagnostics in the future to better characterize the fast electron properties in connection to the lifetime of the filaments (lines 205 - 208 in revised manuscript).

The experimental work is accompanied by PIC simulations which show some similar behavior on shorter timescales, shorter spatial scales and in 2D while also using a very small initial density scalelength.

Indeed, fully-kinetic PIC simulations including realistic target density and collisions have been performed. They capture the development of the instability produced by laser-driven electrons during the first 3 ps, indicating good agreement with the experimental data on the spatial and temporal scales of the filamentary density structures and helping to elucidate the interplay between electromagnetic, space-charge and collisional effects on the growth of the instability. Due to the tremendous computational expense, the simulations are limited to 2D and a few ps. We will further elaborate on how prohibitive 3D simulations are and why the density scalelength used is appropriate in our answers to the comments below.

The data is very nice but there has also been a lot of work previously on this instability in HED plasmas– not all of which has been referenced.

1) Experimental Study of Current Filamentation Instability

B. Allen et al., Phys. Rev. Lett. 109, 185007 (2012)

2) Observations of the filamentation of high-intensity laser-produced electron beams

M. S. Wei et al., Phys. Rev. E 70, 056412 (2004)

3) Weibel-Induced Filamentation during an Ultrafast Laser-Driven Plasma Expansion

K. Quinn et al., Phys. Rev. Lett. 108, 135001 (2012)

4) Dynamics of the Electromagnetic Fields Induced by Fast Electron Propagation in Near-Solid-Density Media L. Romagnani et al. Phys. Rev. Lett. 122, 025001 (2019).

5) Current filamentation instability in laser wakefield accelerators

C M Huntington et al., Phys Rev Lett., 106 :105001 (2011)

We thank the Reviewer for acknowledging the quality of the data presented in the manuscript and for the references on previous studies of filamentation in laser-plasma interactions. We would like to highlight that our experiment is the first to directly image and resolve the

spatio-temporal evolution of the density modulations associated with this instability in solid-density plasmas. Following the Reviewer's suggestion, we have now added the additional references to the manuscript.

Why are the simulations not 3D? It seems that the cylindrical geometry of the wire will play a significant effect on the physics – especially since the wire diameter is about the same as the focal spot diameter.

Unfortunately, it is not possible to model the interaction in 3D for several ps while taking into account the correct solid plasma density, collisions, and the laser finite spot size/focusing effects. The 2D simulations of the laser-wire interaction presented in the manuscript already required ~1 million CPU hours each, modeling ~10 billion particles. A 3D simulation would require ~1 billion CPU hours, which is completely out of reach even for the largest supercomputers.

Despite these computational constraints, we have performed different simulations in order to assess the role of the target cylindrical geometry in the instability formation. In particular, we have performed additional 2D top-view (round-target) simulations with the same target and laser parameters as in the side-view case presented in the manuscript. The results are shown in Fig. 1 of this response below. As we can see, the growth and spatial scale of the density filaments, as well as the peak magnetic field amplification, is very similar to that observed in the planar geometry, with the density perturbation amplitude being ~70% of the value observed in the planar case. In the manuscript, we opted to include the simulation analysis based on the 2D side-view geometry, as that corresponds to the probing geometry in the experiment. We also note that the extent of the filaments inside the target (up to ~5 μm for the copper target) is independent of the target geometry. This will be further discussed in our answer to the next comment.

We have added a statement in the Methods regarding these results ("Particle-In-Cell Simulations section", lines 484 - 489 in the revised manuscript).

Figure 1: a) Electron density map in the top-view simulation and b) in the side-view simulation. c) Lineouts of the electron density (blue), ion density (red) and magnetic field intensity (black) taken at $x = -3\mu\text{m}$ for the top-view simulation and d) for the side-view simulation. The data is taken at $t = 1.1$ ps during the growth of the filaments.

Why do the filaments only extend to the center of the wire? This would be expected if the density modulations observed are due to plasma structures on the outside surface of the wire (i.e. the area that has been irradiated). How do the authors know that the density modulations are not due to such hydrodynamic structure (perhaps induced by pre-pulses) on the irradiated surface of the wire.

The longitudinal extent of the density filaments is indeed a very interesting question that we are actively exploring. We have carefully checked that this is not associated with the target geometry or hydrodynamic structures on the outside of the target surface.

The MEC short-pulse laser possesses very good contrast, with pre-pulses only carrying $\sim 2\mu\text{J}$ (see additional details in the answer to the following comments). This is largely insufficient to produce any significant hydrodynamic modulations of the target before the main pulse arrives.

This is further confirmed by comparing the X-ray images taken right after the main pulse arrival with the cold wire reference images (see Fig. 2 of this response), which show that the target front surface is not modulated and remains sharp for multiple picoseconds. Additionally, we also see that the hydrodynamic evolution of the target surface from the main pulse interaction occurs on much longer time scales at $t > 20$ ps with the propagation of a shock and following the thermal expansion of the target [as observed in Fig. 1 of the manuscript]. As a result, hydrodynamic modulations cannot be responsible for the filamentary structures observed.

Figure 2: X-ray images of (a) the cold (static) wire reference and (b) the laser-driven wire 1.5ps after the interaction. (c) shows the division between the laser-driven image and the cold reference image to reveal the X-ray transmission relative to the static cold wire target (as seen for multiple pump-probe delays in Figure 2 of the manuscript). As can be seen in these images, the front of the target is still sharp at 1.5ps after the laser-target interaction. The edges of the target are visible in c) due to flux variation in the X-ray beam and resulting changes in the phase contrast effects of the very steep density front, leading to slight variations of the fresnel interference pattern in these near-field images. (d) shows the division between two different cold reference images taken of the same wire target that show the same edge fringes to confirm that the edges in (c) are not caused by the laser-target interaction.

It is also important to point out that our X-ray images are most sensitive to the solid density plasmas and sharp gradients therein. Such strong contrasts in density could not be produced in an expanded pre-plasma outside the wire, where the density would be too low for X-rays and the associated density gradients would need to be much sharper to produce similar images—not consistent with hydrodynamic origin.

We also observe that the extent of the filaments can be longer than the wire center and reach close to the full diameter as shown in Fig. 3 of this response for the case of the Al wire with a diameter of 15 micrometer, thus indicating that their extent is not determined by the irradiated area. We note that the extent of the filaments observed in the experiments is also consistent with that observed in our simulations. We have been exploring the simulations and theory further in order to develop a model of how the fast electron density and divergence as well as

the target collisionality control the spatial extent of the filaments that will be left for a separate publication.

Figure 3: X-ray image of a 15 μm aluminum target at $t = 20\text{ps}$. The filaments clearly develop past the center of the wire target.

We have added a section to the Methods addressing the laser-contrast and expected pre-plasma scale length (“Laser contrast and pre-plasma formation, line 321 - 338 in revised manuscript). We have also added a sentence in the main text of the revised manuscript that mentions that the hydrodynamic evolution is evolving on longer time scales and cannot explain the observed density filaments (Lines 148 - 152 in revised manuscript).

The experimental electron temperature is derived from the slope of the proton spectrum. This is problematic since the proton spectra is highly dependent on the laser pulse duration (total pulse energy) rather than the peak intensity. There are many review papers about this.

The Reviewer is correct that for long pulse ($> \text{ps}$) lasers and significant pre-plasmas the proton spectrum can depend strongly on the pulse duration and on the fast electron production in an extended pre-plasma. However, this is not the regime of our experiment.

As we discuss in detail in the next point, in our experiment the level of pre-plasma was low given that the laser pulse intensity contrast was very good: $\sim 10^{-9}$ until -5 ps and $\sim 10^{-7}$ after (see Fig. 4 in our answer to next comment). The total energy in the pre-pulse is of the order of $\sim 2\mu\text{J}$. Given that the pre-pulse arrives only a few picoseconds before the main pulse, there is insufficient time for significant hydrodynamic expansion, and thus the pre-plasma scale length is expected to remain well below $1 \mu\text{m}$. Under similar conditions, previous studies, such as by Malka and Miquel (Phys. Rev. Lett. 77, 75, 1996), as well as recent review papers, for example Rusby et al., Phys. Plasmas 31, 040503 (2024), support the ponderomotive scaling for estimating the hot electron temperature, due to the negligible role of pre-plasma formation.

Furthermore, the pulse duration in our experiments (150 fs) is also short for any significant plasma expansion to occur during the laser-plasma interaction itself. Indeed, we see that our measured proton spectra are in very good agreement with Mora's model for a non-pre-expanded plasma slab (see extended Figure E1 of the manuscript). In order to develop further confidence in our estimate of the fast electron temperature, we compare the derivation from the slope of the proton spectrum with the main theoretical models that have been previously established in the literature (namely, the Wilks' and Haines' models), finding good agreement as well. Finally, the fast electron temperature is also in good agreement with the PIC simulations.

For all these reasons, we are very confident in the fast electron temperature that is derived from the experimental measurements and corresponding theoretical models.

We have revised the manuscript to explicitly mention and discuss the high-contrast of the MEC laser (lines 99 and 127 in revised manuscript) as well as in the Methods section ("Laser contrast and pre-plasma formation, lines 296 and 321 - 338 in revised manuscript), and added a more detailed discussion on the models for the fast electron temperature and their regime of validity in the Methods (lines 364 - 370 in revised manuscript).

What is the laser contrast and the laser and the expected (or measured) density scalelength?

The MEC short-pulse laser achieves high temporal contrast through a pulse cleaning technique based on noncollinear sum-frequency mixing of the signal and idler from an optical parametric amplifier (OPA), as described in [E. Cunningham et al., Appl. Phys. Lett. 114, 221106 (2019)]. As shown in the Tundra autocorrelation trace (Fig. 4 of this response), the contrast reaches $\sim 10^{-9}$ up to -5 ps and $\sim 10^{-7}$ beyond that. A feature at ~ 3 ps is noted but, as mentioned in the publication, it is likely an artifact of the contrast measurement diagnostic. In the following, we include a back-of-the-envelope calculation of the expected pre-plasma scalelength due to this pre-pulse.

Using absorption data from femtosecond laser interactions with solid copper [S. E. Kirkwood et al., Phys. Rev. B 79, 144120 (2009)], significant energy absorption begins around an intensity threshold of $\sim 1 \times 10^{12}$ W/cm². This increase corresponds to changes in electron surface temperature and electron-electron collision frequency. According to the MEC contrast curve, this intensity threshold is crossed at approximately -3 ps. By integrating the laser intensity from -3 ps to -0.5 ps and applying intensity-dependent absorption coefficients for copper from [S. E. Kirkwood et al., Phys. Rev. B 79, 144120 (2009)] ($\sim 5\%$ for $I_L < 10^{12}$ W/cm², $\sim 10\%$ for $10^{12} < I_L < 10^{13}$ W/cm², $\sim 34\%$ for $I_L > 10^{13}$ W/cm²), we estimate that up to ~ 2 μ J is deposited in the target during this pre-pulse window. Beyond -0.5 ps, the contrast measurement is limited by the diagnostic instrument function.

Assuming this energy is deposited and retained within a surface layer defined by a 5 μ m diameter and a depth of ~ 25.8 nm (10.8 nm from optical penetration plus ~ 15 nm from ballistic

hot electron transport [Rethfeld et al., J. Phys. D 50, 193001 (2017)]], we estimate an affected volume of $V = 5.07 \times 10^{-13} \text{ cm}^3$. Using solid copper's atomic density ($n_{\text{Cu}} = 8.45 \times 10^{22} \text{ atoms/cm}^3$), the number of atoms in the volume is $N_i \approx n_{\text{Cu}} \times V = 4.28 \times 10^{10}$. Assuming full deposition of $2 \mu\text{J}$ (E_L) and partial ionization ($Z \approx 4-5$), an upper estimate for the resulting free electron temperature is $T_e = 2/3 \times E_L [\text{eV}] / (Z \times N_i) = 30 - 50 \text{ eV}$. Calculating the resulting plasma expansion velocity, we obtain $v_{\text{exp}} = (ZkT_e/m_{\text{Cu}})^{1/2} \sim 0.02 \mu\text{m/ps}$ ($Z = 5$, $T_e = 50 \text{ eV}$). Since the pre-pulse arrives only a few picoseconds before the main pulse, the resulting pre-plasma does not have time to significantly expand, and the scalelength remains below $0.1 \mu\text{m}$ at the time of interaction.

Based on these calculations, we have also explored the role of pre-plasma in the interaction using PIC simulations. In particular, we have varied the pre-plasma scalelength in the range $0.03-0.12 \mu\text{m}$ and found that it did not significantly affect the fast electron properties and the filamentary structures that develop in the dense plasma.

We have added a statement regarding the pre-pulse energy and laser contrast in the Methods ("Laser contrast and pre-plasma formation, lines 321 - 338 in revised manuscript).

Figure 4: Tundra contrast measurement for the MEC laser [E. Cunningham et al., Appl. Phys. Lett. 114, 221106 (2019)].

Reviewer #3:

The Ms "Time-resolved X-ray imaging of the current filamentation instability in solid density plasmas" reports about imaging laser-matter interaction with very high time resolution using the combination of a FEL and probing laser-matter interaction. The topic of radiography is always an interesting topic within the community as far as it can unlock some bottlenecks within this research area. As such, the paper could be interesting for the community. However, probing of matter using X-rays has already been done in the past, such as using laser-accelerated protons to resolve quick electromagnetic phenomena (e.g. see the PRL by Romagnani 2005). For the moment it is therefore not completely clear, what superiority the quite complex setup used by the author has compared to what has been used in the past and what additional information can be retrieved with this complex setup in addition to what other methods would be able to detect. I think that authors should emphasize this in their Ms such as to convey a clearer message and show superiority of their (quite complex) setup compared to "easier" solutions using only X-rays or laser-accelerated protons.

We thank the Reviewer for the careful consideration and for recognizing the interest of our work. We are happy to further clarify the novelty and importance of our measurements. We must stress that the measurements of the sub-micron density modulations in the solid-density plasma provided in our work are the first of its kind and transform our ability to characterize the evolution of the filamentation instability in these high-energy-density conditions as we will describe in detail below. The use of an XFEL has critical advantages over optical and laser-driven X-ray and proton sources for probing sub-micron modulations (associated with kinetic instabilities) in solid-density plasmas, including:

- 1) **Spatial Resolution:** Kinetic plasma instabilities grow on very small spatial scales which are very challenging to probe at solid density. In the case of the filamentation instability, as we show in our work the most unstable mode has a wavelength of $\sim 1 \mu\text{m}$, which means that to resolve the density modulations we need a source size smaller than 0.5 microns. Unfortunately, most laser-driven proton radiography and X-ray backlighters currently are still limited to source sizes of multiple micrometers [e.g. H.-S. Park et al., *Physics of Plasmas* 15.7, p. 072705 (2008); K. Vaughan et al., *High Energy Density Physics* 9.3, pp. 635–641 (2013); P. K. Singh et al., *Scientific Reports* 12, p. 8100 (2022)], which are not sufficient for resolving the small density filaments. Additionally, laser-generated X-ray sources do not present the required spectral coherence to operate effectively in a microscopy setup as presented in our manuscript. The large bandwidth of these sources would lead to blurring of any features of that size, meaning that even in a setup aimed at compensating for the relatively large source sizes of laser-generated X-ray sources, the required spatial resolution could not be reached. Consequently, the sub-micrometer resolution ($\sim 200\text{nm}$) of the MXI, which is achieved in our experiment due to the brightness (large signal-to-noise), coherence and effective source size of the LCLS is absolutely crucial for measuring these small density fluctuations and at the moment cannot be matched by laser-generated secondary sources;

- 2) **No Deflection by Surface Fields:** Previous studies [e.g., Romagnani et al., Phys. Rev. Lett. 95, 195001 (2004); Quinn, K. et al., Phys. Rev. Lett. 108, 135001 (2012); Albertazzi, B. et al., Rev. Sci. Instrum. 86, 043502 (2015); Ruyer, C. et al., Nature Physics 16, 983–988 (2020)] have used proton radiography to probe the electromagnetic (EM) fields, associated with the filamentary instabilities in laser-plasma interactions. However, these could not resolve the filamentary fields in the solid-density plasma near the interaction region, which is the most relevant to understand the development of the instability and fast particle transport. The reasons for this are both the limited spatial resolution (as argued in the previous point) and the fact that other sources of EM fields in the interaction, including surface fields, will deflect charged particles preventing/corrupting the measurement. For these reasons, these previous studies were limited to probing the fields either in the lower-density expanding plasma or foam or away from the interaction region, where the spatial scale of the filaments becomes of the order to 50 micrometers and can be resolved by laser-driven secondary sources. This underscores the critical importance of X-ray lasers, and of our results in particular, to understand not only the microphysics and nonlinear dynamics of the instability in solid-density plasmas, but also how it will affect particle and energy transport in such dense plasma.
- 3) **Probing Electron Density, Not Fields:** Finally, our measurements are unique in that they probe directly the density and not the fields. It is due to that capability that we were able to demonstrate for the first time that electrostatic effects and ion motion play an important role in the development of the filamentation instability in laser-solid interactions. As argued in our paper, if that was not the case, i.e. for a purely electromagnetic instability, there would be no total density modulations and no filamentary X-ray images would be obtained. Proton radiography is sensitive to electromagnetic fields, rather than electron density, and as such the importance of electrostatic and ion motion effects would not be revealed by proton probing, even if it could deliver the required spatial resolution.

Furthermore, a significant benefit of XFELs lies in reliable, high-repetition rate data acquisition, which during our experiment played a crucial role in generating the high-resolution, clean images presented in the manuscript. These images were only made possible by taking hundreds of reference shots of the X-ray background and the static wire target for effectively removing the X-ray background and revealing small density fluctuations within the target.

For these reasons, these are clearly first-of-a-kind measurements of density fluctuations associated with the current-filamentation instability in solid-density plasmas, which at the moment are only possible with an XFEL through its unique combination of coherence, flux, high-repetition rate and spatial resolution of just a few hundred nanometers that cannot be matched by existing laser-driven sources.

Finally, we would also like to emphasize that the experimental setup is actually not very complex. In fact, following our experiment, several X-ray imaging techniques are now established as standard configurations for experiments at both LCLS and EuXFEL.

We have modified the manuscript to better emphasize the strong relevance of X-ray imaging for investigating solid density laser-plasma interactions (lines 74 - 86 in revised manuscript).

Abstract: authors write "Our findings indicate that magnetic fields on the order of 10 megagauss are produced, which will help confine high-energy electrons within the filaments with important implications for the transport of energetic particles in plasmas."

What about electric fields ? Are they not of importance ? What implications can this have ? What applications have authors in mind that could have an impact on this ?

The electrostatic component of the electric field, produced by space-charge effects, is short-lived as it will serve primarily to excite the ion motion needed to maintain charge neutralization. There will also be an induced electric field (along the filaments) associated with the growth of the magnetic field. This induced field can be estimated as $E \sim \Gamma B \lambda = 0.01 \text{GV/cm}$, where λ is the filament wavelength, B is the magnetic field and Γ is the instability growth rate. This field will last for the duration of the magnetic field amplification phase (~ 1 ps) and is too weak to significantly affect the energy of fast electrons. Thus, the dominant field produced in the dense plasma is the magnetic field as seen in our simulations.

Such strong magnetic fields can have important implications and applications. First of all, they can affect the transport of energetic electrons, since the fields are of the order of the Alfvén limit. This is important for applications such as fast ignition and proton acceleration in laser-solid interactions (e.g. TNSA), since both rely on the efficient transport of fast electrons in the dense target. This fundamental understanding is also relevant for the study of charged-particle (cosmic-ray) transport in astrophysical plasmas, where the filamentation instability is thought to be driven and play an important role but cannot be directly characterized [M. V. Medvedev, ApJ 540 704 (2000); A. Spitkovsky, ApJ 682 L5 (2008)]. These strong magnetic fields could also enable novel radiation sources based on laser-solid interactions, potentially with very high energy and brilliance [A. Benedetti et al., Nature Photonics 12, pages 319–323 (2018); D. J. Stark et al., Phys. Rev. Lett. 116, 185003 2016].

We have added a statement regarding the implications of such strong fields in the conclusion paragraph (lines 276 - 283 in revised manuscript).

Page 3: the resolution seems to be 50 fs and <200 nm. Again, what would be the resolution using laser-accelerated protons (that can probe at ps time frames) or laser-accelerated X-rays, is a resolution of 200 nm required?

We have addressed these requirements in our first response. As we explained above, indeed sub-micron resolution is needed, but this is not the only requirement. To directly probe these

filamentary density structures at solid density we need a combination of penetrating radiation, source size, coherence, flux and bandwidth that at present is not possible with laser-driven secondary sources, underscoring the importance of the measurements provided in our work.

Page 5: it is not completely clear what "The temporal evolution of the filament wavelength" is indicating ? How has this information been retrieved from Fig. 2 ? Maybe a small explanation (arrows etc) could help the reader understand the underlying calculations.

From linear theory, we expect a dominant, well-defined wavelength (the fastest growing) to develop in the density filaments. By analyzing the X-ray images at different times, we observe that this wavelength (the separation between filaments) is changing over time. This is an important nonlinear effect that has been theoretically associated with the merging of filaments via the magnetic force but never verified experimentally. For this reason, we have analyzed how the wavelength of the filaments grows over time, showing that it is consistent with the filament merging model and revealing that ions play a critical role in setting the time for filament merging, which had not been realized before.

The evolution of the filament wavelength was measured using lineouts of the X-ray images taken at different pump-probe delays. Each maximum and minimum of the signal is individually fitted by a Gaussian function and grouped pair-wise. The distance between each pair is then used to extract a distribution of filament wavelengths.

We have improved the manuscript to introduce the methodology used to retrieve the filaments wavelength from the X-ray images, added an arrow in Extended Figure 3 in the revised manuscript and included a reference to the Methods in the main text, where the complete methodology is described (lines 167 - 168).

Page 7: the statement "These experimental results reveal a critical importance of electrostatic effects and ion motion in the dynamics of the asymmetric current filamentation instability" - how is this importance revealed? The ion motion seems to be a new information, how is that information retrieved?

We would like to clarify that in the context of the instability growth, electrostatic effects and ion motion are directly related to each other. For a purely electromagnetic filamentation instability, the total plasma density modulation would be zero, i.e. the density modulation of the fast electrons would be exactly compensated by the density modulation of the background electrons. If that were the case, as considered in most previous theoretical studies of the instability, X-ray images would not show any filamentary density modulation. Thus, by measuring filamentary density modulations we are showing that electrostatic effects are indeed present.

These effects arise due to the density and temperature asymmetry between fast and background electrons, which cause them to pinch in the self-generated magnetic fields at different rates. This causes a charge imbalance and therefore for the instability to grow, background ions will need to move in order to maintain the plasma quasi-neutral. This has very

important consequences for the growth of the instability, as it will significantly slow down its evolution. This is shown in Fig. 4d in the manuscript, where we can see that the growth rate is reduced by more than one order of magnitude by electrostatic effects, and that the resulting growth time (\sim ps) is consistent with the time at which the filaments are observed experimentally.

We further show that ion motion is important in the nonlinear (merging) dynamics of the filaments. As discussed in the paper, if filament merging involved only electrons (as previously considered, e.g. in [M. V. Medvedev et al., ApJ 618 L75 (2005)]), the merging time would be \sim 85 fs (for the copper target), whereas the measured merging time is \sim 11 ps consistent with the ion motion.

We have revised the manuscript (lines 157 - 166, 187 - 188) to better convey this important aspect of our work.

Page 8 (and theory): it would be interesting to compare the PIC results to the experimental results. Authors state that there is a good agreement in the text, but finally there is no figure comparing the results, which would strengthen their statement, even visually.

We thank the referee for this suggestion. Indeed, we have used PIC simulations to investigate the relative contribution of electrostatic effects/ion motion and resistivity, which cannot be easily separated in the experiment. In particular, by comparing the PIC simulation results, the linear theory and the experimental measurements, we can contrast different models describing the microphysics of the filaments in the solid density target. In the manuscript we discuss this comparison and the good agreement obtained in three important metrics:

1. **Filament wavelength.** The filament wavelength measured experimentally as filaments are first observed is \sim 1 micron, as can be seen in Figures 2 and 3 in the manuscript. This is consistent with both the wavelength measured in the PIC simulation (when including collisional effects; Fig. 4b) and theory (Fig. 4d), which is also \sim 1 micron. A visual comparison can already be made between Fig. 2 and Fig. 4b. Following the Reviewer's suggestion, in order to make this comparison clearer, we have added the experimental measurement to Figure 4d. In our opinion, the most important aspect of this comparison is that, in simulation and theory, we show that collisions have a strong impact in determining the filament wavelength and must be taken into account to match the experimentally measured values;
2. **Filamentation growth rate.** We observe experimentally that filaments arise 1.5-2 ps after the laser-target interaction. This time scale is reasonably consistent with the measured growth time of the instability in the simulations, which is 1.3 ps (the inverse of the measured growth rate of 0.8ps^{-1}). While we do not measure a growth rate directly in the experiment, we can compare this with the time at which filaments are first observed. We have added this to Fig. 4d, where the new green triangle marker indicates the time filaments are first observed in the experiments and the corresponding wavelength. We can see that there is good agreement with the simulation results (red marker);

3. **Magnetic field.** In the experiment we estimate the magnetic field amplitude by comparing the temporal evolution of the filament's wavelength with an analytical model for filament merging. From this, we obtain a value of 7-10 MG. These values are comparable to the magnetic field amplitude measured in the simulations, which reach values of 7 MG as seen in Figure 4c (black curve).

We have modified Figure 4d in the revised manuscript to include a data point for the experimentally measured wavelength and time at which the filaments appear in the X-ray images, to put our measurements in perspective with the results obtained from PIC simulations and linear theory (Fig. 4d and associated caption in revised manuscript).

Page 9: Author mention: "Our results reveal the combined importance of space-charge and resistive effects on the growth and nonlinear dynamics of the current filamentation instability." - What does this mean ? Are the two, space-charge and resistive effects, individually important, or only its combination ? What is the impact on the filamentation individually.

This point is discussed between lines 232 and 274 in the revised manuscript, right before the sentence quoted by the Reviewer and illustrated in Fig. 4d. As we can see when comparing the solid (no space-charge) and dashed black (with space-charge) curves in Fig. 4d, space-charge effects will primarily slow down the instability, reducing its growth rate by more than one order of magnitude. Collisional effects on the other hand will primarily shift the most unstable filamentary mode to larger wavelengths, as can be seen by comparing black (no collisions) and red (collisions) circular markers in Fig. 4d. Thus, each of these processes is individually important and to match the experimentally observed growth time and wavelength we need to include both, indicating they both impact the development of the filamentation instability in solid-density plasmas. This is now clearer with the addition of the green marker indicating the experimental measurements in Fig. 4d, allowing for a more direct visual comparison as suggested by the Reviewer.

We would like to highlight that this is an important new result that had not been established experimentally before our work.

In general, the main message of the paper is not fully clear: Initially it seems that the main message is about a new imaging diagnostics ("Time-resolved X-ray imaging"). Then it evolves to the importance of magnetic fields in plasma and finally to filamentation issues, space-charge and resistive effects. I would encourage the authors to find a guideline ("fil conducteur") related to what they want to say within the Ms and build the content/results/discussion accordingly.

We appreciate the Reviewer's suggestion. We note that these aspects are all connected. As discussed in our response, probing the filamentation instability in the interaction region of the intense laser with a solid plasma has been extremely challenging. By using a high-brightness and narrow-bandwidth XFEL we have managed to resolve for the first time the spatio-temporal

evolution of the plasma density filaments in the interaction. This revealed very important aspects for the understanding of this instability that were not established before:

- 1) Space-charge effects and ion motion are important, slowing down the instability by more than one order of magnitude when compared to the purely electromagnetic case;
- 2) Collisional effects are important in determining the wavelength of the filamentation instability at solid-density plasmas;
- 3) Strong magnetic fields, of order 10 MG, are produced in the filamentary region that can affect fast electron transport.

These results can have important implications to applications that range from studying relativistic astrophysical plasma instabilities in the laboratory for the first time to energy transport in inertial confinement fusion plasmas to novel high-energy radiation sources.

We have revised the manuscript, including the abstract (lines 24 - 29 and 32-34 in revised manuscript), introduction (lines 74 - 86 in revised manuscript), and conclusions (lines 276 - 283 in revised manuscript) to make this clearer. We have also added a new paragraph at the end of the introduction (lines 87 - 94 in revised manuscript, before starting discussing the experimental setup) to discuss the innovative aspects of the measurements and the important findings it revealed.

We thank the Reviewers for carefully assessing the revisions to our manuscript and for their positive feedback. Below, we reproduce the Reviewers' comments (in black) and provide our response (in red).

We believe that we addressed all the Reviewers' concerns and our manuscript is now suitable for publication in Nature Communications.

Reviewer #1:

I have read carefully the revised manuscript "Time-resolved X-ray imaging of the current filamentation instability in solid-density plasmas" and the response of author to the referees' remarks. I have found my critics well addressed, as well as I've appreciated the detailed and clear responses to the rigorous remarks of the other referees. In the current form I can recommend the manuscript for publication as an article in Nature Communications.

We thank Reviewer #1 for the encouraging remarks on the revised manuscript.

Reviewer #2:

The authors of "Time-resolved X-ray imaging of the current filamentation instability in solid-density plasmas" have responded to my previous review and have added further explanations and references. However on the whole I do not believe that this paper is suitable for Nature Communications. The diagnostic is very impressive, however the Weibel instability has been explored in many other previous papers.

We appreciate Reviewer #2 for their comments. While there have been previous studies of the Weibel instability, as we discussed in the previous report, none could directly resolve the development of this instability in the solid density plasma region. Indeed the power of a X-ray Free Electron Laser presents a transformative opportunity to probe high energy density plasmas in an unprecedented way as demonstrated in our work. It is also important to make clear that the physics we have uncovered, including the interplay of space charge and collisional effects and the non-linear evolution associated with filament merging, is of primary importance to understand the development and impact of the instability and had long been elusive. Thus, our results provide a significant advance that in our opinion deserves publication in Nature Communications.

In addition there are still many aspects about this explanation that are not convincing including the density of ion perturbations probed, the duration of the filamentary structures, the length of the filaments and the 2D numerical modeling which lasts only 2 psec. The

observed structures are much more likely to be due to hydrodynamics on the wire surface or due to the electrothermal instability as in previous observations.

It would have been helpful if the Reviewer had specified which aspects of our explanation were not convincing. In response to the previous report we have provided a thorough and detailed explanation of these aspects, as also recognized by the other Reviewers, which in our opinion have addressed all the previous points of Reviewer #2. Regarding possible hydrodynamic structures on the wire surface we reiterate that due to the high contrast of the laser there is no time for hydrodynamic processes to develop on picosecond timescales associated with the filamentary structures in our experiment. This is further confirmed by the clear absence of filamentary structures before the arrival of the main pulse. Finally, as also discussed in our response to the previous report, if either hydrodynamic or electrothermal instabilities were to develop in the low-density region outside of the target, as discussed in previous experiments using nanosecond lasers (which have very different conditions from ours), X-ray probing would not be sensitive to these. Thus, these are not consistent with our observations.

Reviewer #3:

The authors made significant efforts for clarifying the concerns of the reviewers and the paper is now much more robust and understandable. Well done.

From my side, the logical sequence is still difficult to follow: the paper being very long (10 page), a small introductory sentence guiding the reader in the "Result" section would help, but this is up to the authors.

We thank the Reviewer for their positive comments and their suggestion. Based on the previous report we had already extended the paragraph before the "Results" section to guide the reader on the main findings of our work. Following the Reviewer's suggestion we have now added subsections to the "Results and Discussion" section to further help the reader through the structure of our manuscript.